# PoNet: Pooling Network for Efficient Token Mixing in Long Sequences

**Chao-Hong Tan**[1]*, **Qian Chen**[2], **Wen Wang**[2], **Qinglin Zhang**[2], **Siqi Zheng**[2], **Zhen-Hua Ling**[1]

[1]National Engineering Laboratory for Speech and Language Information Processing,
University of Science and Technology of China
[2]Speech Lab, Alibaba Group
[1]`chtan@mail.ustc.edu.cn, zhling@ustc.edu.cn`
[2]`{tanqing.cq, w.wang, qinglin.zql, zsq174630}@alibaba-inc.com`

## Abstract

Transformer-based models have achieved great success in various NLP, vision, and speech tasks. However, the core of Transformer, the self-attention mechanism, has a quadratic time and memory complexity with respect to the sequence length, which hinders applications of Transformer-based models to long sequences. Many approaches have been proposed to mitigate this problem, such as sparse attention mechanisms, low-rank matrix approximations and scalable kernels, and token mixing alternatives to self-attention. We propose a novel Pooling Network (**PoNet**) for token mixing in long sequences with linear complexity. We design multi-granularity pooling and pooling fusion to capture different levels of contextual information and combine their interactions with tokens. On the Long Range Arena benchmark, PoNet significantly outperforms Transformer and achieves competitive accuracy, while being only slightly slower than the fastest model, FNet, across all sequence lengths measured on GPUs. We also conduct systematic studies on the transfer learning capability of PoNet and observe that PoNet achieves 95.7% of the accuracy of BERT on the GLUE benchmark, outperforming FNet by 4.5% relative. Comprehensive ablation analysis demonstrates effectiveness of the designed multi-granularity pooling and pooling fusion for token mixing in long sequences and efficacy of the designed pre-training tasks for PoNet to learn transferable contextualized language representations. Our implementation is available at `https://github.com/lxchtan/PoNet`.

## 1 Introduction

Transformer (Vaswani et al., 2017) has become the state-of-the-art (SOTA) architecture for sequence modeling in a wide variety of fields, including natural language processing (NLP), computer vision, speech processing, applications to genomics data, etc. The key reason for Transformer's success is its self-attention mechanism, which computes dot-product between input representations for each pair of positions in the full sequence. Proved to be greatly effective in learning contextualized representations, Transformer becomes the backbone for dominant pre-trained language models (PLM) in NLP, such as BERT (Devlin et al., 2019) and RoBERTa (Liu et al., 2019). These PLMs demonstrate strong transfer learning capabilities and have achieved SOTA widely on NLP tasks. However, self-attention has quadratic time and memory complexity to the input sequence length (Vaswani et al., 2017), which becomes the bottleneck for applying the vanilla Transformer to long sequence modeling tasks and for scaling up Transformer-based PLMs.

A broad spectrum of approaches have been proposed to address the efficiency problem of self-attention, as summarized in (Tay et al., 2020). One major direction approximates the dense full self-attention using techniques such as introducing sparsity (Child et al., 2019; Beltagy et al., 2020; Zaheer et al., 2020; Zhang et al., 2021a; Yang et al., 2021; Zhang et al., 2021b), low-rank approximations of the softmax attention matrix (Katharopoulos et al., 2020; Wang et al., 2020a; Zhu & Soricut, 2021; Xiong et al., 2021; Peng et al., 2021; Choromanski et al., 2021), and locality

---
*Work is done during the internship at Speech Lab, Alibaba Group.

sensitive hashing (Kitaev et al., 2020). These approximation approaches exploit observations that token interactions have strong locality, hence the importance and in turn the attention should decrease with the increase of the distance between a query token and a key token. Several of these works achieve $O(N)$ theoretical complexity. However, these models often require selecting relatively large local regions for fine-grained attention in order to approximate full self-attention, hence the scaling constants hidden by $O(N)$ are often large and hinder significant improvements in speed and memory usage. It is observed that performance of the approximation approaches is usually inversely related to their speed (Beltagy et al., 2020; Zaheer et al., 2020). Another major direction replaces the self-attention structure with more efficient structures, such as MLP-Mixer (Tolstikhin et al., 2021), FNet (Lee-Thorp et al., 2021), AFT (Zhai et al., 2021), and Fastformer (Wu et al., 2021). Despite significant accelerations gained by efficient transformers, they are rarely evaluated both on effectiveness of the inductive bias and the transfer learning capability, except BigBird, FNet, and Nyströmformer[1] as we understand. BigBird-Base (Zaheer et al., 2020) outperforms BERT-Base on the GLUE benchmark (Wang et al., 2019) without significant speedup (slowdown for input length $< 3K$) (Tay et al., 2021). FNet-Base trains 80% faster than BERT on GPUs at 512 input lengths but only achieves 92% of BERT-Base's accuracy (Lee-Thorp et al., 2021).

In this work, we propose a novel Pooling Network (**PoNet**), aiming to simultaneously advance long sequence modeling capacity and transfer learning capabilities while improving speed and memory efficiency. PoNet replaces the $O(N^2)$ complexity self-attention with a $O(N)$ complexity multi-granularity pooling block. We design multi-granularity pooling and pooling fusion to model different levels of token interactions. Multi-granularity pooling incorporates three types of pooling, from coarse-grained to fine-grained, in each sublayer. Global aggregation aggregates information of the entire sequence into a single token. Segment max-pooling captures the paragraph or sentence level information. Local max-pooling captures the more important local information. These three poolings are fused to produce the output feature of the multi-granularity pooling block. Then through the residual connection, this output feature is further aggregated into each token.

The contributions of this paper are summarized as follows:

- We propose a novel PoNet architecture as a drop-in replacement for self-attention in Transformer, achieving linear time and memory complexity. We propose multi-granularity pooling and pooling fusion to capture different levels of contextual information and comprehensively model token interactions. To the best of our knowledge, our work is the first to explore the full potential of the simple pooling mechanism for token mixing and modeling long-range dependencies.
- Extensive evaluations show that PoNet achieves competitive performance on the Long Range Arena benchmark (Tay et al., 2021) and significantly outperforms Transformer by +2.28 absolute (+3.9% relative) on accuracy, with efficiency up to 9 times faster and 10 times smaller than Transformer on GPU. Also, PoNet demonstrates competitive transfer learning capabilities, with PoNet-Base reaching 95.7% of the accuracy of BERT-Base on the GLUE benchmark. Ablation analysis further proves effectiveness of designed multi-granularity pooling and pre-training tasks.

## 2 RELATED WORK

**Efficient Transformer Variants** Among the models to approximate full self-attention, Longformer (Beltagy et al., 2020) ($O(N)$) sparsifies the full self-attention into three attention patterns of sliding window, dilated sliding window, and global attention. BigBird (Zaheer et al., 2020) ($O(N)$) combines global attention, local attention, and random attention. Poolingformer (Zhang et al., 2021a) ($O(N)$) uses a two-level attention schema, with the first level using a smaller sliding window to aggregate local information and the second level using a larger window with pooling attention to reduce time and memory cost. Focal Transformer (Yang et al., 2021) ($O(N)$) uses both fine-grained local interactions and coarse-grained global interactions to balance efficiency and effectiveness of capturing short- and long-range dependencies. Transformer-LS (Zhu et al., 2021) ($O(N)$) approximates the full attention by aggregating long-range attention via dynamic projections and short-term attention via segment-wise sliding window. H-Transformer-1D (Zhu & Soricut, 2021) ($O(N)$) exploits a matrix structure similar to Hierarchical Matrix. AdaMRA (Zhang et al., 2021b) ($O(N)$) leverages a multi-resolution multi-head attention mechanism and kernel attention. Luna (Ma et al., 2021) ($O(N)$) introduces an additional fixed length sequence served as query to

---

[1]Nyströmformer was evaluated only on subsets of GLUE (Xiong et al., 2021).

attend to the original input while the output is served as key and value to attend to the original input. Apart from the sparse attention models, other approximation approaches explore locality sensitive hashing and matrix approximation methods. Reformer (Kitaev et al., 2020) $O(NlogN)$ replaces self-attention with locality sensitive hashing. Performer (Choromanski et al., 2020; 2021) ($O(N)$) approximates softmax attention by leveraging random features. Linformer (Wang et al., 2020a) approximates the self-attention matrix with a low-rank factorization. Nyströmformer (Xiong et al., 2021) ($O(N)$) approximates the softmax attention with the Nyström method by sampling a subset of columns and rows. However, these approximation methods have strengths on certain tasks and may cause accuracy degradation on many other tasks.

Our work is in another line of research on replacing self-attention with more efficient token mixing mechanisms. MLP-Mixer (Tolstikhin et al., 2021) ($O(N^2)$) applies two separate linear transformations on the hidden state dimension and the sequence dimension. FNet (Lee-Thorp et al., 2021) ($O(NlogN)$) replaces the self-attention sublayer with 2D-FFT mixing sublayer. AFT-local/conv (Zhai et al., 2021) ($O(sN), s < N$) first combines the key and value with a set of learned position biases and then combines the query with this result via element-wise multiplication. Fastformer (Wu et al., 2021) ($O(N)$) first models global context via additive attention then models interactions between global context and input representations through element-wise product. Shared Workspace (Goyal et al., 2021) ($O(N)$) proposes the idea of using a shared bottleneck to tackle the problem of quadratic dependence in attention.

**Pre-training Tasks**  It has been observed that both underlying model architecture and pre-training are crucial to performance of PLMs. BERT (Devlin et al., 2019) with a Transformer encoder is pre-trained with masked language modeling (MLM) and next sentence prediction (NSP) tasks on large-scale unlabeled text corpora including the English Wikipedia and BooksCorpus. MLM predicts the masked token from context. NSP predicts whether a sentence pair is contiguous or not in the original source. Many approaches are proposed to improve these two tasks and show that more challenging pre-training tasks may help PLMs learn better and more transferable language representations.

Whole word masking (WWM) (Devlin et al., 2019; Cui et al., 2019) and SpanBERT (Joshi et al., 2020) outperform BERT on many tasks. WWM simultaneously masks all WordPiece tokens belonging to the same word and forces the model to recover a complete whole word. SpanBERT randomly samples contiguous spans inside of individual tokens and augments MLM with a new task to predict the entire masked span. RoBERTa (Liu et al., 2019) reports ineffectiveness of NSP and removes it from pre-training. ALBERT (Lan et al., 2020) replaces NSP with a sentence-order prediction (SOP) task to predict whether two consecutive sentences are in the right order or not, for learning fine-grained inter-sentence coherence. StructBERT (Wang et al., 2020b) extends SOP to a new sentence structural objective (SSO) as a ternary classification on two sentences $(S_1, S_2)$ to decide whether $S_1$ precedes or follows $S_2$ or the two sentences are noncontiguous. More challenging tasks for learning inter-sentence relations and document/discourse structures (Iter et al., 2020; Lee et al., 2020; Ding et al., 2021) show promising performance improvements on PLMs.

## 3  MODEL

Our work is inspired by the External Attention (EA) approach proposed in (Guo et al., 2021) for visual tasks. An input sequence of tokens $\boldsymbol{x} = \{x_1, ..., x_N\}$ are mapped to an embedding matrix denoted by $\boldsymbol{H}(\in \mathbb{R}^{N \times d}) = \{\boldsymbol{h}_1, ..., \boldsymbol{h}_N\}$, where $N$ is the sequence length and $d$ is the hidden dimension. EA uses two linear layers to implement external and shared memories, which facilitates learning correlations across all samples and hence serves strong regularization to and improves generalization of the attention mechanism with linear complexity. We simplify EA into multi-layer perceptron and $softmax$, and observe that by infusing the sequence-level information into each token through the denominator term $\sum_{n=1}^{N} e^{h_n}$, $softmax$ provides context modeling capabilities. However, $softmax$ involves calculations of exponents, which is still slow. Consequently, we consider using pooling as an alternative to capture contextual information with significantly reduced complexity. We propose a Pooling Network (**PoNet**) as a drop-in replacement for the self-attention sublayer in Transformer, as shown in Figure 1. PoNet models different levels of contextual information through a multi-granularity pooling (MP) block consisting of three components, namely, global aggregation (GA), segment max-pooling (SMP), and local max-pooling (LMP). These pooling features are then aggregated through pooling fusion.

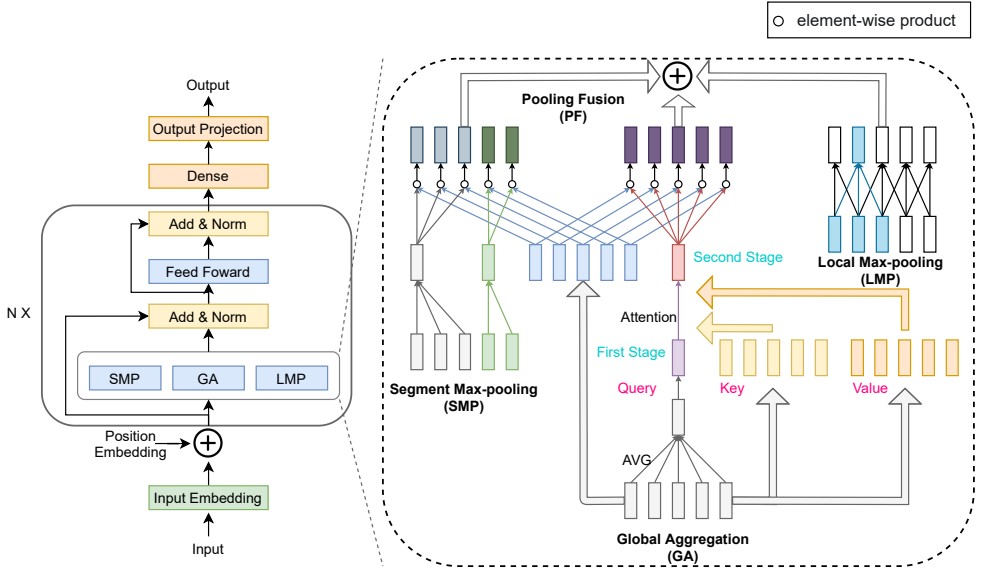

Figure 1: The illustration of the PoNet model architecture. The right enlarged view shows multi-granularity pooling (GA, SMP, LMP) and pooling fusion (Section 3).

## 3.1 Multi-granularity Pooling

In order to capture contextual information at different levels, we design multi-granularity pooling. First, different linear projections are applied on the input for poolings at different granularities:

$$\boldsymbol{H}_* = \boldsymbol{H}\boldsymbol{W}_* + \boldsymbol{b}_*, \tag{1}$$

where $*$ represents $Q_g, K_g, V_g$ for GA, $s$ for SMP, $l$ for LMP, and $o$ for pooling fusion, in total six $\boldsymbol{W}_*$. $\boldsymbol{W}_* \in \mathbb{R}^{d \times d}$ and $\boldsymbol{b}_* \in \mathbb{R}^d$ are parameters to be learned. The different $H_*$ are then used for different poolings.

### 3.1.1 Global Aggregation

A Global Aggregation (GA) module is carefully designed aiming to both capture the most important global information for each token and also to guarantee an overall linear computational complexity. We calculate the first stage value $\boldsymbol{g}$ for GA by averaging at the sequence level:

$$\boldsymbol{g} = \frac{1}{N}\sum_{n=1}^{N} \boldsymbol{h}_{Q_g\, n} \in \mathbb{R}^d, \tag{2}$$

Note that $\boldsymbol{g}$ is only a rough representation of the sequence. Beltagy et al. (2020); Zaheer et al. (2020) introduced a global attention mechanism, which adds a set of global tokens with randomly initialized values to always attend to the whole sequence. Inspired by these works, we perform cross-attention on the first stage value for GA. The first stage value $\boldsymbol{g}$ is used to perform a query on the input sequence to compute the second stage value $\boldsymbol{g}'$ for GA, as follows:

$$\boldsymbol{g}' = Attention\{\boldsymbol{g}, \boldsymbol{H}_{K_g}, \boldsymbol{H}_{V_g}\}. \tag{3}$$

The cross-attention on $\boldsymbol{g}$ enables each token to attend to the whole sequence and hence the resulting second stage value $\boldsymbol{g}'$ provides a more accurate sequence representation, compared to $\boldsymbol{g}$. Note that since $\boldsymbol{g}$ is a single token with the length equaling 1, the computational complexity of attention in Eq. 3 is $O(N)$. Theoretically, using average-pooling for generating $\boldsymbol{g}$, the rough representation of the sequence, could keep more information for the next step cross-attention and generating the second stage value for GA, instead of discarding most information and only focusing on the most salient features as the max-pooling function does. This hypothesis is verified as we observe a better model performance from using average-pooling for generating $\boldsymbol{g}$ than max-pooling. In contrast, the outputs of SMP and LMP are directly used as final representations, hence we choose max-pooling for SMP and LMP, which is also empirically verified to produce a better model performance.

### 3.1.2 SEGMENT MAX-POOLING

The information loss from compressing a long sequence into a single global token could be enormous and hence become detrimental to the sequence modeling capacity. We introduce an intermediate level between tokens and the global token by segmenting an input sequence into segments, exploring prior knowledge of structure in the data when available, and introduce Segment Max-pooling (SMP). The use of structural information for segmentation is adjustable for different tasks, which is described in detail in Section 4. As explained earlier, max-pooling is performed on each segment at each dimension $j \in [1, d]$, as follows, where $K$ denotes the number of segments:

$$s_j^k = \max\{h_{s0j}^k, h_{s1j}^k, ..., h_{sN_kj}^k\}, \tag{4}$$

$$\boldsymbol{s}^k = \{s_1^k, ..., s_d^k\} \in \mathbb{R}^d, \tag{5}$$

$$\boldsymbol{S} = \{\boldsymbol{s}^1, ..., \boldsymbol{s}^K\} \in \mathbb{R}^{K \times d} \tag{6}$$

### 3.1.3 LOCAL MAX-POOLING

Many prior works (Zhu & Soricut, 2021; Yang et al., 2021) demonstrate the crucial role of capturing local information for the sequence modeling capabilities. We introduce Local Max-pooling (LMP), designed as a standard max-pooling over sliding windows, to capture contextual information from neighboring tokens for each token. Different from GA and SMP, the window for LMP is overlapped. Similar to GA, LMP is also applied at the sequence level and the left and right boundaries of the input sequence are padded to ensure that the output length equals the original input length. The LMP values $\boldsymbol{L} \in \mathbb{R}^{N \times d}$ are computed. The size and stride of the sliding window are set to 3 and 1 in all our experiments unless stated otherwise.

### 3.1.4 POOLING FUSION

First, we model the interactions between GA and each token by computing $\boldsymbol{G}_n$ through the element-wise product between the second stage value of GA $\boldsymbol{g}'$ and each token, as follows:

$$\boldsymbol{G}_n = \boldsymbol{g}' \circ \boldsymbol{H}_{on}, \tag{7}$$

The reason for conducting Eq. 7 instead of directly using $\boldsymbol{g}'$ as the output is to avoid all tokens from sharing the same global token. Otherwise, it will cause the token representations to converge to the same global token in the subsequent addition fusion layer. This effect, in turn, will make the token representations become more homogeneous, and consequently degrade performance on tasks such as sentence-pair classifications. The SMP token $\boldsymbol{S}$ is shared by all tokens in each segment. Hence, for the same rationale of mixing the global token with each token, the same operation is conducted to mix the SMP token with each token and to compute $\boldsymbol{S}'$ as:

$$\boldsymbol{S}'_n = \boldsymbol{S}_{k(n)} \circ \boldsymbol{H}_{on}, \tag{8}$$

where $k(n)$ denotes the segment index of the $n$-th token. The above three features are added up as the final output of our multi-granularity pooling block, as illustrated in Figure 1:

$$\boldsymbol{P} = \boldsymbol{G} + \boldsymbol{S}' + \boldsymbol{L}, \tag{9}$$

where $\boldsymbol{P}$ is used to replace the original self-attention output of Transformer. We compare PoNet to related models, including Fastformer (Wu et al., 2021), Luna (Ma et al., 2021), Longformer (Beltagy et al., 2020) and BigBird (Zaheer et al., 2020), in detail in Appendix D.

### 3.2 COMPLEXITY ANALYSIS

We only analyze the computational complexity of the proposed multi-granularity pooling block, since we only replace the self-attention sublayer in Transformer with this block and keep other modules in Transformer unchanged. The six $\boldsymbol{HW}_*$ in Eq. 1, where $*$ represents $Q_g, K_g, V_g$ for GA, $s$ for SMP, $l$ for LMP, and $o$ for pooling fusion, require $6Nd^2$ computations. GA requires $2Nd$ ops (Eq. 3), SMP and LMP have no matrix multiplication, and Pooling Fusion requires $2Nd$ ops (Eq. 7 and Eq. 8). Hence, the total number of multiplication ops is $6Nd^2 + 4Nd$. We further simplify computations. By switching the order of Eq. 1 and Eq. 2 into first performing the average pooling then the affine function, computation can be reduced from $Nd^2$ to $d^2$. Adding $\boldsymbol{g}'$ and $\boldsymbol{S}$

| Model | ListOps(2K) | Text(4K) | Retrieval(4K) | Image(1K) | Pathfinder(1K) | AVG. |
|---|---|---|---|---|---|---|
| Transformer(1) | **36.37** | 64.27 | 57.46 | 42.44 | 71.40 | 54.39 |
| Longformer (1) | 35.63 | 62.85 | 56.89 | 42.22 | 69.71 | 53.46 |
| BigBird (1) | 36.05 | 64.02 | **59.29** | 40.83 | 74.87 | **55.01** |
| Performer (1) | 18.01 | **65.40** | 53.82 | **42.77** | **77.05** | 51.41 |
| Transformer(2) | **36.06** | 61.54 | **59.67** | **41.51** | 80.38 | **55.83** |
| Linear (2) | 33.75 | 53.35 | 58.95 | 41.04 | **83.69** | 54.16 |
| FNet (2) | 35.33 | **65.11** | 59.61 | 38.67 | 77.80 | 55.30 |
| Transformer(3) | 37.10 | 65.02 | 79.35 | 38.20 | **74.16** | 58.77 |
| Performer(3) | 18.80 | 63.81 | 78.62 | 37.07 | 69.87 | 53.63 |
| Reformer(3) | 19.05 | 64.88 | 78.64 | 43.29 | 69.36 | 55.04 |
| Linformer(3) | 37.25 | 55.91 | 79.37 | 37.84 | 67.60 | 55.59 |
| Nyströmformer(3) | 37.15 | 65.52 | 79.56 | 41.58 | 70.94 | 58.95 |
| FNet | 37.40 | 62.52 | 76.94 | 35.55 | FAIL | 53.10 |
| PoNet (Ours) | **37.80** | **69.82** | **80.35** | **46.88** | 70.39 | **61.05** |

Table 1: Results on the Long Range Arena (LRA) benchmark (AVG: average accuracy across all tasks). Results with (1) are cited from (Tay et al., 2021), with (2) are from (Lee-Thorp et al., 2021), with (3) are from (Xiong et al., 2021). We implement our PoNet and re-implement FNet based on the PyTorch codebase from (Xiong et al., 2021) and use the same experimental configurations to ensure a fair comparison. For each group, the best result for each task and AVG are bold-faced.

first and then performing element-wise product can reduce $2Nd$ to $Nd$, compared to conducting Eq. 7 and Eq. 8 separately. After the simplifications, the total number of multiplication ops is $(5N + 1)d^2 + 3Nd$. The multi-granularity pooling block hence has linear time and memory complexity with respect to the input length.

## 4 EXPERIMENTS

We first evaluate PoNet on the Long Range Arena (LRA) benchmark (Tay et al., 2021) and compare PoNet to the vanilla Transformer and a series of efficient transformer variants on accuracy, training speed, and memory usage. Next, we study the transfer learning capability of PoNet in the commonly used paradigm of pre-training followed by fine-tuning. We evaluate the fine-tuning performance on the GLUE benchmark (Wang et al., 2019) as well as a set of long-text classification tasks. All baseline models and PoNet use the same "Base" model configuration as BERT-Base (Devlin et al., 2019). PoNet-Base has 124M parameters (see the first paragraph in Appendix A for more details). More experimental details, results and analyses, and comparisons to other models are in Appendices.

### 4.1 LONG-RANGE ARENA BENCHMARK

**Comparison on Accuracy** The LRA benchmark is designed to assess the general capabilities of capturing long-range dependencies. LRA consists of six tasks spanning structural reasoning (*ListOps*), similarity reasoning (Byte-level *Text* classification and document *Retrieval*, *Image* classification), and visual-spatial reasoning (*Pathfinder*). We use the PyTorch codebase from (Xiong et al., 2021)[2] to implement FNet and our PoNet and evaluate on LRA after first replicating the results from Nyströmformer in (Xiong et al., 2021). We keep the same hyperparameter setting as used by (Xiong et al., 2021) for all of our LRA evaluations and report results on five tasks in Table 1[3]. All the baseline models are summarized in Section 2 and the **Linear** variant (Lee-Thorp et al., 2021) denotes replacing the self-attention sublayer with two linear projections, one applied to the hidden dimension and one applied to the sequence dimension. Note that due to different code implementations, results for same models could differ across groups in Table 1 (see Appendix A.1 for details). It is important to point out that all models in the third group are implemented with the same codebase from (Xiong et al., 2021) and the same experimental configurations to ensure a fair comparison within this group. As shown in Table 1, compared to the first and second groups and the cited results in the third group marked with (3), PoNet achieves competitive performance on LRA. PoNet outperforms the vanilla Transformer by **+2.28** (**61.05** over 58.77) and Nyströmformer by +2.10 on the average accuracy and consistently accomplishes better performance on all tasks

---

[2]https://github.com/mlpen/Nystromformer

[3]We exclude the Path-X task since all the evaluated models failed in the Path-X task, probably due to its very long 16K sequence length (Tay et al., 2021)

| Seq. length | 512 | 1024 | 2048 | 4096 | 8192 | 16384 |
|---|---|---|---|---|---|---|
| | Training Speed (steps/s)↑ | | | | | |
| Transformer | 45.1 | 19.4 | 6.3 | 1.8 | OOM | OOM |
| Performer | 39.4(0.9x) | 25.0(1.3x) | 14.3(2.3x) | 7.8(4.3x) | 4.0 | 2.0 |
| Nyströmformer | 39.1(0.9x) | 30.3(1.6x) | 20.0(3.2x) | 11.5(6.4x) | 6.1 | 3.1 |
| FNet | **83.4(1.8x)** | **61.3(3.1x)** | **38.1(6.0x)** | **21.4(11.9x)** | **11.0** | **5.4** |
| PoNet (Ours) | 50.4(1.1x) | 40.1(2.1x) | 27.8(4.4x) | 16.2(9.0x) | 8.7 | 4.5 |
| | Peak Memory Usage (GB)↓ | | | | | |
| Transformer | 1.4 | 2.5 | 6.7 | 23.8 | OOM | OOM |
| Performer | 1.5(1.1x) | 2.1(0.8x) | 3.1(0.5x) | 5.4(0.2x) | 9.8 | 18.7 |
| Nyströmformer | 1.2(0.8x) | 1.5(0.6x) | 1.9(0.3x) | 2.8(0.1x) | 4.5 | 8.2 |
| FNet | **1.1(0.8x)** | **1.2(0.5x)** | **1.4(0.2x)** | **1.7(0.1x)** | **2.3** | **3.8** |
| PoNet (Ours) | **1.1(0.8x)** | 1.3(0.5x) | 1.7(0.2x) | 2.4(0.1x) | 3.6 | 6.5 |

Table 2: Comparison of GPU training speed (in steps/s, the higher the better) and peak memory consumption (in GB, the lower the better) on various input sequence lengths on the LRA text classification task (using the same hyper-parameter setting for this task as in (Xiong et al., 2021), with speed-up and memory-saving multipliers relative to Transformer shown in parentheses. The best results are bold-faced with the second-best results underlined.

compared to Transformer, Performer, Reformer, Linformer, FNet, and Nyströmformer, except slightly weaker than Transformer on the Pathfinder task. It is reasonable to conclude that PoNet outperforms BigBird on LRA since the margin +2.28 from PoNet over Transformer in the third group is significantly larger than the margin +0.62 from BigBird over Transformer in the first group. To the best of our knowledge, PoNet achieves very competitive accuracy on LRA against Transformer and recent efficient transformers, only lower than 63.09 from AdaMRA (Zhang et al., 2021b)[4], 61.95 from Luna-256 (Ma et al., 2021)[5], and 61.41 from H-Transformer-1D (Zhu & Soricut, 2021).

**Comparison on Speed and Memory Consumption** Table 2 compares the GPU training speed and peak memory consumption of PoNet to Transformer, Performer, Nyströmformer, and FNet on a single NVIDIA V100 chip, on input sequence lengths from 512 up to 16384. We observe that PoNet is the second fastest model and consumes the second smallest memory footprint in the group, consistently on all sequence lengths, much faster than Transformer, Performer, and Nyströmformer and lighter than them, and only slightly slower and heavier than FNet. Also, the speedup from PoNet over Transformer escalates on longer input sequence lengths.

## 4.2 TRANSFER LEARNING

The paradigm of pre-training followed by fine-tuning has been extensively applied and accomplished SOTA results in a wide variety of NLP tasks. Therefore, it is critical to evaluate the transferability of PoNet. We perform pre-training on PoNet and evaluate the fine-tuning performance on the GLUE benchmark and a set of long-text classification benchmarks.

To facilitate a fair comparison on the transfer learning ability, we pre-train BERT, FNet, and PoNet with the same MLM (Devlin et al., 2019) and sentence structural objective (SSO) as in StructBERT (Wang et al., 2020b) on the English Wikitext-103 (100M words) and BooksCorpus (800M words) datasets[6] (in total 5GB data). The total pre-training loss is $\mathcal{L} = \mathcal{L}_{MLM} + \mathcal{L}_{SSO}$. All three models are base-uncased and pre-trained using the same configuration (Appendix A.2). Figure 2 illustrates validation accuracy of the same MLM and SSO tasks from BERT, FNet, and PoNet pre-training. MLM accuracy from PoNet is only slightly worse than that from BERT while the gap on SSO accuracy between them is a bit larger. PoNet achieves significantly better MLM and SSO accuracy than FNet, consistent with its better sequence modeling capability shown on LRA.

---

[4]AdaMRA (Zhang et al., 2021b) also re-implemented BigBird with the same codebase from Xiong et al. (2021) and reported AVG 59.43 from BigBird, which is worse than PoNet.

[5]Luna-256 (Ma et al., 2021) outperforms their reimplemented Transfomer by +2.71, while PoNet achieves +2.28 gain over Transformer implemented based on the same codebase and using the same configurations.

[6]https://huggingface.co/datasets

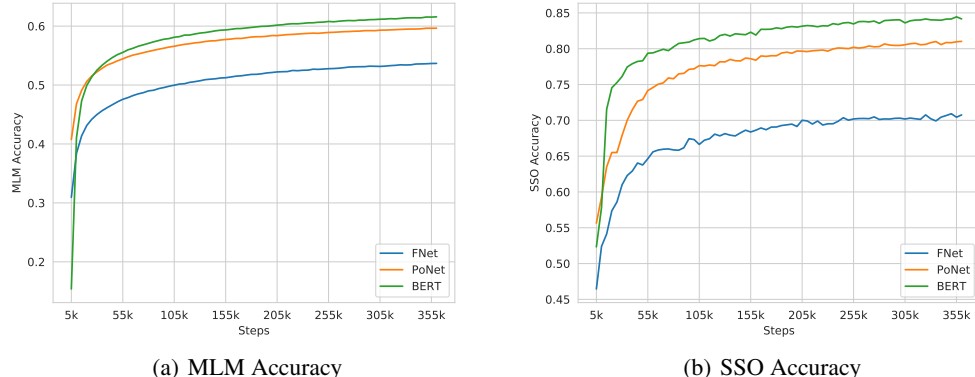

(a) MLM Accuracy          (b) SSO Accuracy

Figure 2: MLM and SSO validation accuracy against the numbers of training steps from BERT-Base, FNet-Base, and PoNet-Base. All models are uncased.

| Model | MNLI(m/mm) | QQP | QNLI | SST-2 | CoLA | STS-B | MRPC | RTE | AVG. |
|-------|-----------|-----|------|-------|------|-------|------|-----|------|
| BERT-Base | 81.35/80.98 | 88.89 | 88.01 | 91.17 | 47.66 | 87.83 | 86.66 | 69.31 | **80.21** |
| FNet-Base | 73.13/73.66 | 85.75 | 80.50 | 88.65 | 40.61 | 80.62 | 80.84 | 57.40 | 73.46 |
| PoNet-Base (Ours) | 76.99/77.21 | 87.55 | 84.33 | 89.22 | 45.36 | 84.57 | 81.76 | 64.26 | 76.80 |

Table 3: GLUE **Validation** results from PoNet, BERT, and FNet. All models are uncased and pre-trained with the same configurations (Appendix A.2) with **340K** steps. We report the best GLUE results for each model from multiple hyper-parameter configurations (Appendix A.3). We report the mean of accuracy and F1 for QQP and MRPC, matthews correlations for CoLA, spearman correlations for STS-B, and accuracy for other tasks. MNLI(m/mm) means match/mismatch splits.

**Results on GLUE** The GLUE benchmark covers a diverse range of challenging natural language understanding tasks and is widely adopted for evaluating transfer learning models. The tasks can be split into two groups, single-sentence tasks including CoLA and SST-2, and sentence-pair tasks including MRPC, QQP, STS-B, MNLI, QNLI, and RTE[7]. The special token "[SEP]" is used as the segment separator. For fine-tuning PoNet on GLUE, inputs to single-sentence tasks include three segments as "[CLS]" Sentence-1 "[SEP]"; whereas inputs to sentence-pair tasks include five segments "[CLS]" Sentence-1 "[SEP]" Sentence-2 "[SEP]". These segments are used for computing SMP (Section 3). Table 3 shows the results for the best base learning rate (no early stopping) on the GLUE **Validation** split (see Appendix A.3 for more details), providing a fair comparison since all three models are pre-trained and fine-tuned with the same pre-training data (5GB data)/tasks/hyper-parameters with 340K steps. Table 3 shows that PoNet achieves 76.80 AVG score, reaching **95.7%** of the accuracy of BERT on GLUE (80.21) and outperforming FNet by **4.5%** relatively. These performance comparisons are consistent with the pre-training accuracies shown in Figure 2. The results also prove that PoNet is equally competitive in both single-sentence and sentence-pair tasks. When pre-trained on the same 16GB data used for the official BERT pre-training with MLM+SSO tasks up to 1M steps, PoNet also achieves **95.9%** of BERT's accuracy on GLUE (see Appendix B.2).

**Results on Long-text Classification** We also evaluate the fine-tuning performance of the pre-trained PoNet on four long-text classification datasets, including Hyperpartisan news detection (HND) (Kiesel et al., 2019)[8], IMDb (Maas et al., 2011), Yelp-5 (Zhang et al., 2015), and Arxiv-11 (He et al., 2019). As can be seen from Table 4, PoNet-Base outperforms BERT-Base on HND (+8.2 $F_1$) and Arxiv (+0.75 $F_1$) and reaches 99% of BERT-Base's $F_1$ on IMDb and Yelp-5.

## 5 ABLATION ANALYSIS

We conduct ablation analysis on contributions from multi-granularity pooling and pre-training tasks. By applying leave-one-out on PoNet components, we create variants as PoNet w/o Second Stage

---
[7]Following (Devlin et al., 2019; Lee-Thorp et al., 2021), we exclude WNLI.
[8]We use the train/validation/test division provided by Beltagy et al. (2020).

| Model | HND($F_1$) | IMDb($F_1$/Acc) | Yelp-5($F_1$) | Arxiv($F_1$) |
|---|---|---|---|---|
| #Example (#Classes) | 500 (2) | 25000 (2) | 650000 (5) | 30043 (11) |
| #Wordpieces avg. (95thpctl.) | 734 (1,974) | 312 (805) | 179 (498) | 16,210 (32,247) |
| RoBERTa-Base (Zaheer et al., 2020) | 87.8 | 95.3/95.0 | 71.75 | 87.42 |
| Longformer (Beltagy et al., 2020) | 94.8 | **95.7**/−− | −− | −− |
| BigBird (Zaheer et al., 2020) | 92.2 | −−/**95.2** | **72.16** | **92.31** |
| BERT-Base | 88.0 | 94.1/94.1 | 69.59 | 85.36 |
| FNet-Base | 86.3 | 90.4/90.5 | 65.49 | 79.90 |
| PoNet-Base (Ours) | **96.2** | 93.0/93.0 | 69.13 | 86.11 |

Table 4: Fine-tuning results (in $F_1$ and Acc) on long-text classification datasets. For the third group of results, we use the official checkpoints of BERT-Base and FNet-Base (see Appendix A.4).

| Model | Pre-trained tasks | | Downstream tasks | |
|---|---|---|---|---|
| | MLM | SST | CoLA | STS-B |
| PoNet($340K$ steps) | 59.44 | 80.75 | 45.36 | 84.57 |
| PoNet w/o SS-GA | 59.33 | 76.92 | 46.18 | 78.38 |
| PoNet w/o GA | 56.64 | 74.36 | 49.51 | 64.61 |
| PoNet w/o SMP | 56.96 | 78.41 | 44.21 | 84.89 |
| PoNet w/o LMP | 56.53 | 80.27 | 41.44 | 85.55 |
| PoNet w/o (SMP&LMP) | 43.61 | 76.72 | 11.36 | 84.93 |
| PoNet using $\mathcal{L}_{MN}$ | 62.53 | 79.28 | 50.91 | 75.32 |
| PoNet using $\mathcal{L}_{OM}$ | 63.11 | −− | 51.26 | 69.83 |

Table 5: Results of ablation study as accuracy for pre-training MLM and SST (Sentence Structure Task) tasks, matthews correlations for CoLA, and spearman correlations for STS-B. SST denotes NSP when using $\mathcal{L}_{MN}$ and the SSO task otherwise. All pre-training experiments run $340K$ steps.

GA (SS-GA), w/o GA, w/o SMP, w/o LMP, and w/o (SMP&LMP). We also pre-train PoNet with two weaker losses, as $\mathcal{L}_{MN} = \mathcal{L}_{MLM} + \mathcal{L}_{NSP}$ and $Ls_{OM} = \mathcal{L}_{MLM}$, where $\mathcal{L}_{NSP}$ is the NSP loss in BERT. We report pre-training task validation accuracy and GLUE validation scores on the single-sentence CoLA and sentence-pair STS-B tasks in Table 5. Since [CLS] from PoNet w/o GA cannot capture information of the whole sequence, as using [CLS] for classification fails on SST (Sentence Structure Task), CoLA, and STS-B, we use max-pooling of sequence for classification.

Removing GA (PoNet w/o SS-GA, PoNet w/o GA) significantly degrades accuracy on SST and STS-B, showing that sentence-pair tasks heavily rely on the global information. MLM accuracy degrades only slightly from PoNet w/o SS-GA but more significantly from PoNet w/o GA, indicating that MLM also depends on the global information, but when the rough global information is available (PoNet w/o SS-GA), SMP can compensate sufficiently and PoNet w/o SS-GA improves CoLA. Removing GA enhances SMP and LMP learning thus improves CoLA since CoLA relies on SMP and LMP much more than GA. Different from the conclusions on BERT (Devlin et al., 2019; Liu et al., 2019), we find fine-tuning performance of PoNet on sentence-pair tasks highly relies on SST pre-training tasks. Weakening SST loss ($\mathcal{L}_{MN}$, $\mathcal{L}_{OM}$) weakens GA representation learning while enhancing SMP and LMP learning, causing a significant degradation in STS-B accuracy but a significant gain in CoLA. Similarly, removing SMP or LMP enhances GA representation learning and hence improves STS-B accuracy while degrading CoLA accuracy. PoNet w/o SMP&LMP (i.e. GA only) shows a drastic degradation on MLM and CoLA accuracy (45.36 to 11.36). These results confirm that all three poolings are important for the modeling capabilities of PoNet.

## 6 CONCLUSION

We propose a novel Pooling Network (PoNet) to replace self-attention with a multi-granularity pooling block, which captures different levels of contextual information and combines them for a comprehensive modeling of token interactions. Extensive evaluations demonstrate that PoNet achieves both competitive long-range dependency modeling capacity and strong transfer learning capabilities, with linear time and memory complexity. Future work includes further optimization of model structure as well as applying PoNet to a broader range of tasks including generation tasks.

## REPRODUCIBILITY STATEMENT

All data used in the experiments in this paper are open source. Readers can refer to the original papers for details of the datasets, which are cited in our paper. Information on data access can be found in Appendix A. Experimental details are also described in Appendix A. Our implementation is available at `https://github.com/lxchtan/PoNet`.

## ACKNOWLEDGEMENTS

This work was supported by Alibaba Group through Alibaba Research Intern Program.

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

# Appendices

## A   EXPERIMENT DETAILS

**Model Size** For all experiments for PoNet in this paper, parameters $\boldsymbol{W}_{K_g}, \boldsymbol{b}_{K_g}$ and $\boldsymbol{W}_{V_g}, \boldsymbol{b}_{V_g}$ in Equation 1 are shared to reduce the calculations and we observe no performance degradation. After this sharing of $\boldsymbol{W}_{K_g}, \boldsymbol{b}_{K_g}$ and $\boldsymbol{W}_{V_g}, \boldsymbol{b}_{V_g}$ in Equation 1, PoNet-Base has 124M parameters. This is comparable to the 110M parameters for BERT-Base but still a bit larger in the number of parameters. Our ideas for further reducing the number of parameters are summarized in Appendix F.1.

### A.1   LONG-RANGE ARENA BENCHMARK EXPERIMENTAL DETAILS

**Implementations and Hyperparameters** We use the Pytorch codebase from Xiong et al. (2021)[9] to implement our PoNet and re-implement FNet and conduct all LRA evaluations to facilitate a fair comparison. We use exactly the same experimental configurations provided by Xiong et al. (2021)[9]. Note that due to different code implementations, as shown in Table 1, the results from our re-implemented FNet achieve 53.10 average score, lower than 55.30 reported in the original FNet paper (Lee-Thorp et al., 2021) where the original FNet is implemented in JAX/Flax. The Pytorch implementation also causes some difference in results for other models compared to their LRA results reported in the original LRA paper (Tay et al., 2021), which are implemented in JAX/Flax.

For each task, the input sequence is truncated evenly into $K$ segments (as the $K$ in Equation 4). We find that 32 segments for the Image task, 64 segments for the ListOps and Retrieval tasks, 2048 segments for the Text task produce the best results. However, PoNet fails on the Pathfinder task under all three segment configurations. Hence we remain computing SMP on the whole sequence (i.e., only 1 segment) for the Pathfinder task.

**Additional LRA Results** The standard deviations from three runs of our PoNet model on LRA for each task are: ListOps (3.98e-3), Text (4.09e-3), Retrieval (5.89e-3), Image (2.50e-3), and PathFinder (3.2e-3). The standard deviations from three runs of our FNet implementation on LRA for each task are: ListOps (4.66e-3), Text (3.25e-3), Retrieval (2.99e-3), Image (1.57e-2), and all the three runs on the Pathfinder task are failed.

Note that we use the PyTorch codebase from Xiong et al. (2021) to implement FNet and our reported FNet results on LRA in Table 1 and Table 2 are based on our FNet implementation. This implementation adds a Linear layer after the attention, as done for the BERT series but not in the original FNet paper Lee-Thorp et al. (2021) nor in its official Google research code and the HuggingFace implementation. We experiment with removing this Linear layer in FNet and notice a significant performance degradation on LRA from FNet, with the AVG score dropping from 53.10 in Table 1 to 50.65.

**Speed and Memory Comparison** The training speed and peak memory consumption comparisons are conducted on the LRA text classification task on a single NVIDIA Tesla V100 GPU. The input sequence lengths are set from 512 to 16384. Note that since the original LRA github[10] also provides the full length datasets, hence we could truncate the sequences in the LRA text classification task into 8K and 16K lengths. The hyper-parameters are the same as in (Xiong et al., 2021), that is, the hidden size is set to 64, the intermediate size is set to 128, the number of attention heads is set to 2, the number of layers is set to 2, and the batch size is set to 32.

### A.2   PRE-TRAINING DETAILS

For feasible experiment turn-around with our computation resources, we first use the English Wikitext-103 (100M words) [11] and the BooksCorpus (800M words)[12] datasets for pre-training,

---

[9]https://github.com/mlpen/Nystromformer
[10]https://github.com/google-research/long-range-arena
[11]https://huggingface.co/datasets/wikitext
[12]https://huggingface.co/datasets/bookcorpus

in total 5GB data size. For PoNet, natural paragraph segmentations in the datasets, which are marked by "\n", are treated as segments for the SMP computation. For the MLM task, the masking probability is set to 15%. 80% of the masked positions are replaced by "[MASK]", 10% are replaced by randomly sampled words, and the remaining 10% are unchanged. For the SSO task, a long sequence containing several paragraphs is truncated into two subsequences at random positions, with 1/3 probability of replacing one of the subsequences with another randomly selected subsequence, 1/3 probability of swapping the two subsequences, and 1/3 probability unchanged. These three cases are assigned three different labels for the ternary classification. All input sequences are truncated to a maximum sequence length of 512, and to accommodate sentences of different lengths, some input sequences are truncated shorter with a probability of 0.1. The datasets were duped 5 times to alleviate overfitting of SSO tasks, which were also applied by Devlin et al. (2019)[13]. Since the selection of masked positions and sentence pairs are done according to probabilities, we obtain 5 times more training sample pairs. The pre-training implementation is based on the Pytorch codebase from Wolf et al. (2020), with the hyper-parameters shown in Table 6. Each pre-training experiment is run on 4 NVIDIA Tesla V100 GPUs and takes about 9 days.

| | Pre-training | GLUE | Long-text Tasks |
|---|---|---|---|
| Max Steps | $750K$ | – | – |
| Max Epochs | – | 4 | 10 or $2^{\text{I}}$ |
| Learning Rate | 1e-4 | {3e-5, 5e-5, 1e-4, 3e-4} | {3e-5, 5e-5} |
| Batch Size | 192 | {128, 64, 32, 16, 8} | 32 |
| Warm-up Steps | $5K$ | 0 | 0 |
| Sequence Length | 512 | 128 | 4096 |
| Learning Rate Decay | | Linear | |
| Adam $\epsilon$ | | 1e-8 | |
| Adam $(\beta_1, \beta_2)$ | | (0.9, 0.999) | |
| Clip Norm | | 1.0 | |
| Dropout | | 0.1 | |

   [I] The value 2 is used for the high-resource task Yelp-5, and 10 for the other tasks.

Table 6: Detailed hyperparameter settings for the pre-training and fine-tuning experiments. For rows with a single hyperparameter value, the value is used across pre-training and fine-tuning on GLUE and long-text classification tasks.

### A.3 FINE-TUNING ON THE GLUE BENCHMARK

The GLUE datasets[14] can be accessed from Lhoest et al. (2021). The special token "[SEP]" is used as the segment separator. Inputs to single-sentence tasks are processed as "[CLS] S [SEP]"; whereas inputs to sentence-pair tasks as "[CLS] S1 [SEP] S2 [SEP]". We implement all fine-tuning code based on the Pytorch codebase from Wolf et al. (2020). We run 20 sets of hyper-parameter configurations based on Table 6 for fine-tuning BERT, FNet, and our PoNet and report the best GLUE results in Table 3. Note that Table 3 provides a fair comparison between BERT, FNet, and PoNet as all three models are pre-trained with the same pre-training data of English Wikitext-103 (100M words) and BooksCorpus (800M words) datasets[15], the same pre-training tasks of MLM+SSO, the same pre-training configurations (Appendix A.2) (all trained with **340K** steps), and fine-tuned on GLUE with the same configurations.

---

[13]https://github.com/google-research/bert
[14]https://huggingface.co/datasets/glue
[15]https://huggingface.co/datasets

| Model | MNLI(m/mm) | QQP | QNLI | SST-2 | CoLA | STS-B | MRPC | RTE | AVG. |
|---|---|---|---|---|---|---|---|---|---|
| BERT-Base(1) | 84/81 | 87 | 91 | 93 | 73 | 89 | 83 | 83 | 83.3 |
| Linear-Base(1) | 74/75 | 84 | 80 | 94 | 67 | 67 | 83 | 69 | 77.0 |
| FNet-Base(1) | 72/73 | 83 | 80 | 95 | 69 | 79 | 76 | 63 | 76.7 |
| BERT-Base(2) | 85/85 | 89.77 | 91.78 | 92.66 | 58.88 | 89.28 | 89.31 | 70.76 | **83.52** |
| FNet-Base(2) | 75/76 | 86.72 | 83.23 | 90.13 | 35.37 | 81.43 | 80.34 | 59.92 | 74.23 |
| PoNet-Base(Ours)(2) | 78/78 | 87.76 | 85.17 | 89.00 | 47.24 | 85.86 | 83.39 | 63.53 | 77.54 |
| BERT-Base(3) | 83/83 | 89.48 | 90.65 | 91.74 | 51.19 | 89.28 | 88.73 | 67.51 | **81.63** |
| FNet-Base(3) | 75/76 | 86.17 | 82.52 | 88.42 | 40.57 | 83.64 | 80.90 | 61.73 | 74.99 |
| PoNet-Base(Ours)(3) | 79/78 | 87.92 | 86.31 | 89.79 | 45.18 | 87.17 | 84.27 | 66.43 | 78.29 |

Table 7: Extra GLUE **Validation** results. BERT-Base and PoNet-Base are uncased whereas FNet-Base is cased (since the official FNet checkpoint is cased). We report the mean of accuracy and F1 scores for QQP and MRPC, matthews correlations for CoLA, spearman correlations for STS-B, and accuracy scores for other tasks. The MNLI(m/mm) means the match/mismatch splits. Results with (1) are from (Lee-Thorp et al., 2021). Results with (2) and (3) are the best results from searching 20 sets of hyper-parameter configurations based on Table 6 for fine-tuning the pre-trained models. For BERT-Base(2) and FNet-Base(2), we use the official checkpoints provided by authors while for PoNet-Base(2), we pre-train the PoNet model on 5GB data (Wikitext-103 and BooksCorpus). For a fair comparison on model capacity by pre-training with more data, BERT-Base(3), FNet-Base(3), and PoNet-Base(3) are all pre-trained on the same 16GB Wikipedia and BooksCorpus data used for pre-training official BERT checkpoints, trained with MLM+SSO tasks for 1M steps.

## A.4    Fine-tuning on the Long-text Classification Tasks

The HND dataset can be acquired followed the guide in (Beltagy et al., 2020)[16]. The IMDb dataset[17] and Yelp-5 dataset[18] are from Lhoest et al. (2021). The Arxiv-11 dataset is from He et al. (2019)[19].

The max sequence length for all long-text classification experiments is 4096. Since there is no natural paragraph segmentation for these data, we use the NLTK toolkit (Bird et al., 2009) to segment the input into sentences for SMP computation. Note that our PoNet model was pre-trained with max sequence length 512, to be able to fine-tune on 4096 input lengths, following Beltagy et al. (2020), we add extra position embeddings initialized by copying the pre-trained 512 position embeddings recurrently. We implement all the fine-tuning code based on the Pytorch codebase from Wolf et al. (2020) with the hyper-parameters shown in Table 6.

For BERT-Base, we fine-tune using the HuggingFace BERT-Base-uncased checkpoints[20] (pre-trained on Wikipedia and BooksCorpus, in total 16GB data size). We fine-tune FNet-Base (Lee-Thorp et al., 2021) (pre-trained on 700GB C4 data) by converting the official FNet checkpoints using the tool from https://github.com/erksch/fnet-pytorch to be loadable by the Pytorch codebase for fine-tuning. For PoNet-Base, we fine-tune our pre-trained PoNet-Base-uncased with pre-training hyper-parameters shown in Table 6, that is, PoNet-Base pre-trained with **750K** steps.

# B    Additional GLUE Results

## B.1    Compare to Official BERT and FNet Checkpoints

Earlier studies show that due to its model capacity, Transformer-based PLMs benefit from larger pre-training data Liu et al. (2019). Hence, we also show additional GLUE validation results. First, we compare the performance between PoNet pre-trained on 5GB data and the official checkpoints of BERT pre-trained on 16GB Wikipedia and BooksCorpus data and FNet pre-trained on 700GB C4 data, as shown in Table 7. The results with (1) in Table 7 are from (Lee-Thorp et al., 2021). The results with (2) in Table 7 show the best GLUE results from searching 20 sets of hyper-parameter

---

[16]https://github.com/allenai/longformer/blob/classification/scripts/hp_preprocess.py

[17]https://huggingface.co/datasets/imdb

[18]https://huggingface.co/datasets/yelp_review_full

[19]https://github.com/LiqunW/Long-document-dataset

[20]https://huggingface.co/bert-base-uncased

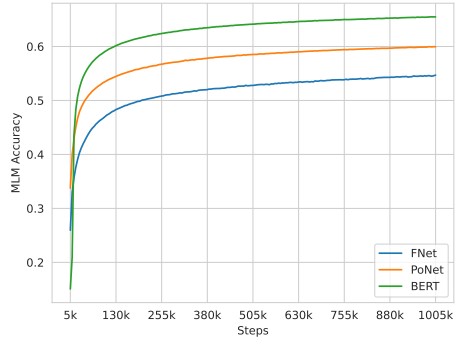

(a) MLM Accuracy on 16GB Datasets

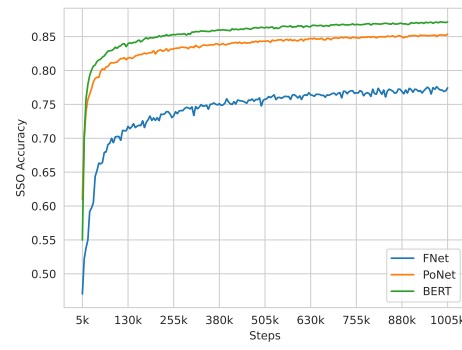

(b) SSO Accuracy on 16GB Datasets

Figure 3: MLM and SSO validation accuracy on 16GB datasets (Wikipedia and BooksCorpus) against the numbers of training steps from BERT-Base, FNet-Base, and PoNet-Base. All models are uncased.

configurations based on Table 6 for fine-tuning the pre-trained models. For BERT-Base(2) and FNet-Base(2), we use their respective pre-trained checkpoints provided by authors. For BERT-Base(2), we fine-tune on GLUE using the Huggingface BERT-Base-uncased checkpoints[20]. For FNet-Base(2), we re-evaluate FNet-Base-cased (Lee-Thorp et al., 2021) on GLUE by converting the official FNet-Base-cased checkpoints[21] using the tool from https://github.com/erksch/fnet-pytorch to be loadable by the PyTorch codebase for fine-tuning. For PoNet-Base(2), we fine-tune our pre-trained PoNet-Base-uncased with pre-training hyper-parameters shown in Table 6 and pre-trained with **750K** steps.

It is important to point out that the GLUE results from official checkpoints of BERT-Base and FNet-Base and our pre-trained PoNet-Base are not comparable since the pre-training data and pre-training tasks are different. BERT-Base was pre-trained on English Wikipedia (2.5B words) +BooksCorpus (800M words) with MLM+NSP, in total 20GB data size. FNet-Base was pre-trained on C4 ( 700GB data size) with MLM+NSP. In contrast, our PoNet-Base was pre-trained on Wikitext-103 (100M words)+BooksCorpus (800M words) with MLM+SSO, in total only 5GB data size. Nevertheless, with the much smaller pre-training data, PoNet-Base still achieves GLUE AVG 77.54, reaching 92.84% of the accuracy of BERT on GLUE (83.52) and outperforms FNet (74.23) by 4.5% relatively, and also better than 76.7 reported in the original FNet paper (they used different code implemented with JAX/Flax).

We also compare other reported results to verify the correctness of our fine-tuning implementations. Note that as reported in Table 7, fine-tuning the FNet-Base-cased official checkpoints in our PyTorch-implemented fine-tuning code achieved 74.23 AVG score on the GLUE Validation set. HuggingFace reproduced FNet-Base-cased with PyTorch and after pre-training FNet-base using the same pre-training data C4 and MLM and NSP tasks as the original FNet paper (Lee-Thorp et al., 2021) and conducting GLUE fine-tuning, their FNet-Base-cased achieved 74.89 AVG score on the GLUE validation set[22], which is comparable to 74.23 from our FNet-Base-cased fine-tuning results. These results verified the correctness of our FNet fine-tuning implementation. HuggingFace also compares the GLUE validation results between their FNet-Base re-implemented in PyTorch and the original FNet implemented in JAX/Flax, and also shows a significant gap from the PyTorch implementations[23]. Hence, we think the difference between our FNet-Base GLUE score and the score reported in the original FNet paper (our 74.23 versus their 76.7) is also due to different code implementation (PyTorch versus JAX/Flax) and different hyper-parameter settings. For the official BERT-Base-uncased checkpoints, we obtain the best GLUE validation results as AVG 83.52, which

---

[21]https://github.com/google-research/google-research/tree/master/f_net

[22]https://huggingface.co/google/fnet-base. Note that we exclude their WNLI accuracy from computing the AVG score for GLUE, following the common practice from (Devlin et al., 2019).

[23]https://huggingface.co/google/fnet-base. HuggingFace shows that the average score on 9 tasks (including WNLI) from FNet-base(PyTorch) is 72.7 while it is 76.7 from FNet-base(Flax official).

is better than the 82.62 AVG score HuggingFace reported from BERT-Base-cased[24]. These results verified the correctness of our BERT fine-tuning implementations.

## B.2 RESULTS FROM USING MORE PRE-TRAINING DATA

In order to investigate the model capacity of PoNet, we also pre-train PoNet on the sufficiently large Wikipedia and BookCorpus dataset used for pre-training the official BERT model, in total 16GB data size. For a fair comparison, we pre-train BERT-Base, FNet-Base, and PoNet-Base with the same 16GB data, the same joint MLM+SSO tasks, up to 1M steps considering the larger data amount. The hyper-parameters for pre-training on 16GB data are the same as shown in Table 6 except for training 1M steps.

The MLM and SSO validation accuracies against the numbers of training steps are shown in Figure 3. Compared to the observations when pre-training BERT-Base, FNet-Base, and PoNet-Base on the 5GB data set, the SSO accuracy from PoNet is only slightly worse than BERT whereas the gap on the MLM accuracy is a bit larger. PoNet achieves significantly better MLM and SSO accuracy than FNet, consistent with its better sequence modeling capability shown on LRA.

GLUE validation results from BERT-Base, FNet-Base, and our PoNet-Base models pre-trained on the same 16GB data with MLM+SSO tasks up to 1M steps are shown as the group (3) in Table 7. Compared with PoNet-Base(2) (pre-trained on 5GB data), PoNet-Base(3) improves the average score of GLUE by a margin of 0.75 (77.54 to 78.29), which proves the potential of our model to capture more knowledge with a larger amount of pre-training data. Note that table 3 shows the GLUE fine-tune results when BERT-Base, FNet-Base, and PoNet-Base are all pre-trained on 5GB data with MLM+SSO tasks, where PoNet-Base reaches **95.7%** of the accuracy of BERT (76.80 against 80.21). It is encouraging to observe here that in this fair comparison with all three models pre-trained on the much larger 16GB data, PoNet-Base(3) also reaches **95.9%** of the accuracy of BERT-Base(3) on GLUE (78.29 against 81.63), demonstrating the competitive transfer learning capabilities of PoNet. Note that BERT-Base(3) has a lower performance than the official BERT-Base(2), which is mainly due to the difference in batch size constrained by machine resource limitations. Although we conduct 1M training steps, the batch size is still smaller than that of the official BERT (98,304 versus 128,000 words/batch). This degradation is also caused by the data pre-processing scripts we use as well as slightly by the FP16 training strategy we adopt. Note that BERT, FNet, and our PoNet all suffer some performance degradation from these factors, and the performance comparison in the group (3) in Table 7 is a fair comparison.

## C  VISUALIZATION

### C.1  STATISTICAL ANALYSIS

To study the importance of the three granularities of pooling, i.e., GA, LMP, and SMP, at different layers of PoNet, we take 4000 examples, calculate the mean of the $L_2$-Norm of each type of pooling for the 4000 examples at each layer, as well as the average of the means for the three types of pooling. The resulting curves are shown in figure 4. Specifically, the norm is calculated as follows,

$$L_2\text{-Norm} = \sum_{h,l,i} \frac{\text{LN}_{h,l,i}}{H * L * I}, \tag{10}$$

$$\text{LN}_{h,l,i} = \sqrt{\sum_d \frac{h_{i,h,l,d}^2}{D}}, \tag{11}$$

where $i, h, l, d$ denote the dim of example, attention head, text length, and hidden unit, respectively. We observe that

(1) All three curves corresponding to the three types of pooling are close to the average curve in the graph, indicating that all three types of pooling play a significant role in PoNet.

---

[24]https://huggingface.co/google/fnet-base

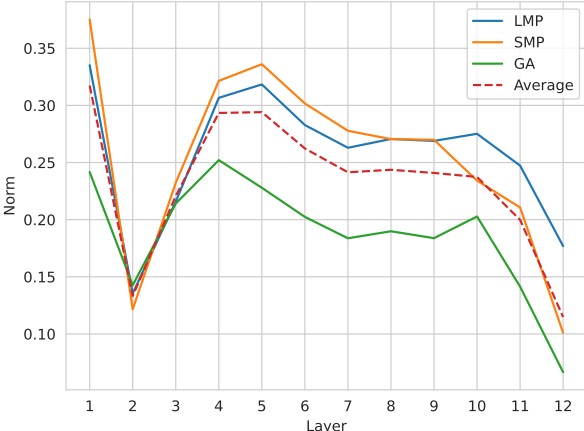

Figure 4: The $L_2$-Norm of the three pooing GA, SMP, and LMP and their average at different layers of PoNet.

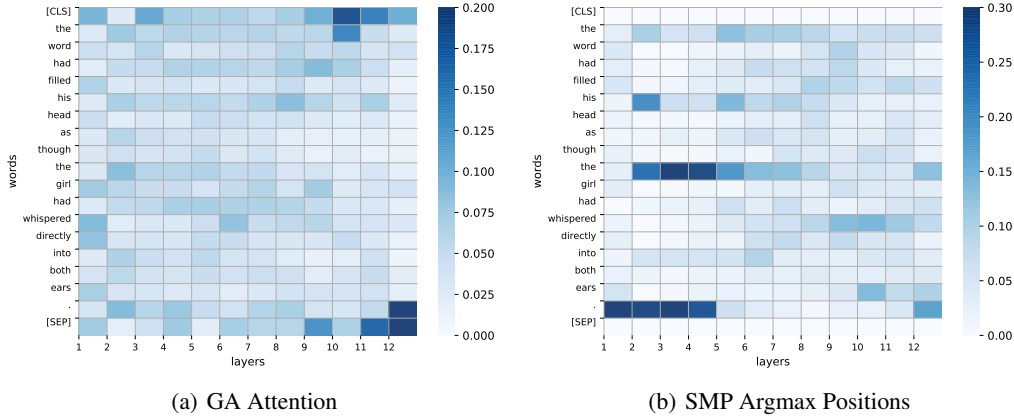

| (a) GA Attention | (b) SMP Argmax Positions |

Figure 5: The GA attention map and SMP argmax positions for the example "The word had filled his head as though the girl had whispered directly into both ears."

This observation is consistent with the ablation results shown in Table 5, which shows that removing GA, removing SMP, or removing LMP in PoNet all significantly degrade the performance on downstream single-sentence and sentence-pair tasks.

(2) All three curves corresponding to GA, SMP, and LMP share a similar trend of going down, up, down, slowly rising and going down. From Layer 1 to 8, the norm values of SMP are greater than those of LMP and GA. Higher than Layer 9, the norm values of LMP turn out to be greater than those of SMP. These observations indicate that the significance of different granularities of pooling changes across layers.

(3) The norm value of GA is relatively low compared to that of SMP and LMP. On the other hand, as shown in the ablation results in Table 5, removing GA from PoNet (PoNet w/o GA) degrades performance on SST and STS-B significantly, demonstrating that the sentence-pair tasks heavily rely on the global information.

## C.2 CASE STUDY

To analyze how PoNet works, we loaded a pre-trained PoNet and selected an example, "The word had filled his head as though the girl had whispered directly into both ears.", for visualization. For

GA, we average the attention weights of all heads as the attention weights of each layer. For SMP, we count the number of times all hidden layer dimensions were taken as the segment maximum value. The resulting GA attention map and SMP argmax positions across layers are shown in Figure 5. The tokens "[CLS]" and "[SEP]" are excluded in the SMP visual map since they belong to a single word segment. To sharpen the display, the maxima in the figure are truncated to 0.2 or 0.3 accordingly. On the bottom layer, we observe that GA first focuses on some of the more important words, such as "whispered" at Layer 1 and then attends to the rest words at Layer 2. This complementary attention allows the model to capture more comprehensive information about the sequences after multiple layers. We observe that SMP is more inclined to capture information about punctuation and pronouns at the bottom level and then some keywords, e.g., "whispered", also begin to receive attention at the higher level, especially at Layer 9-11 based on the SMP argmax positions.

# D    COMPARISON BETWEEN PONET AND OTHER MODELS

## D.1    DIFFERENCE BETWEEN PONET AND FASTFORMER

First, our PoNet consists of three components, i.e., global aggregation(GA), segment max-pooling(SMP), and local max-pooling(LMP). The importance of SMP and LMP is demonstrated in Section 5. Second, the GA component in PoNet is different from Fastformer (Wu et al., 2021) due to different motivations and implementations, as follows.

In the motivation aspect, our GA is inspired by the global attention in Longformer and BigBird, but is also different from them (see Appendix D.2). In contrast, Fastformer uses the additive attention mechanism to model global contexts.

In the aspect of implementation, firstly, comparing GA in PoNet with Fastformer, the global query is computed differently. In Fastformer, the global query vector is a weighted sum of all query vectors; whereas, in PoNet, it is an average pooling of all query vectors.

Secondly, the global query is mixed with key and value in different ways. In Fastformer, element-wise product is performed between the global query vector $q$ and each key vector $k_i$ to obtain a set of vectors $p_i$. A set of learnable vectors are introduced for additive attention to compute weights for $p_i$. After softmax normalization of the weights, the sum of weighted $p_i$ is the global key (GK). Then, element-wise product is performed between $GK$ and each value vector $v_i$ to obtain a set of vectors $U$. After that, $U$ passes through a linear transformation layer to obtain its hidden representations $R$. And the hidden representations $R$ are element-wise summed with $Q$ to obtain the final output of Fastformer.

In our GA, as described in Section 3.1.1, the first stage value for GA, denoted by $g$ in the paper, serves as a query token to interact with the input sequence through cross-attention. There are several differences from Fastformer. First, standard scaled dot-production is performed on $Q$ and $K$ matrices, and the weight of the scalar is obtained directly. Then, softmax is applied on the scaled dot-product results as weights for $V$. The weighted sum of $V$ is computed as the global representation, denoted by $g'$ in the paper, and there is no process of global key ($GK$). Finally, as described in Section 3.1.4, we perform element-wise product between $g'$ and $H_{on}$ to compute the final global representation for pooling fusion, where $H_{on}$ is from projecting input embedding of each token through another linear transformation specific for pooling fusion. Different from Fastformer, there is no linear transformation layer nor summation with $Q$.

The above comparisons show that the GA in our PoNet is quite different from Fastformer. In addition to algorithm comparisons, we also experimented with replacing the GA module in our PoNet with Fastformer, and observed that after the same pre-training and fine-tuning configurations, replacing GA with Fastformer has lower performance compared to our PoNet: On CoLA the performance dropped from 45.36 to 43.28; on STS-B, the performance dropped from 84.57 to 84.46.

## D.2    DIFFERENCE BETWEEN PONET AND LONGFORMER/BIGBIRD

Our design of the multi-granularity pooling and pooling fusion is inspired by previous works such as Longformer and BigBird. Our design considers the importance of capturing context information

from different granularities and combining the captured information. However, our design is different from previous works.

Longformer uses a window-based self-attention to capture local context and a task-specific global attention to encode inductive bias about the task. For example, for classification tasks the global attention attends to the [CLS] token and for question-answering tasks, the global attention attends to all question tokens. In contrast, GA in PoNet is task-agnostic.

BigBird combines global, windowed local, and random attentions. BigBird chooses a subset from existing tokens as global tokens or adds additional global tokens such as [CLS]. These global tokens attend to all existing tokens. The GA in PoNet is computed differently from BigBird (see Section 3.1.1 and Section 3.1.4 for details). Also, PoNet uses GA, segment max-pooling and local max-pooling. We experimented with replacing our windowed local max-pooling with random pooling in an early PoNet structure with a small number of pre-training steps, and observed that this change weakened PoNet's capability of capturing local context and significantly degraded the accuracy of downstream single-sentence CoLA task (from 42.57 to 31.44) while causing a slight gain on the sentence-pair STS-B task (from 80.29 to 80.48). Hence, we did not use random attention in the final structure of PoNet.

### D.3 DIFFERENCE BETWEEN PONET AND LUNA

The mechanism in Luna (Ma et al., 2021) is also somewhat similar to our GA module. Luna introduces an additional "global" sequence of fixed length $L$, which is used as a query to cross-attend with the original input to obtain a global representation of length $L$. Then, it is served as key and value to perform the second cross-attention with the original input to obtain the final output. In fact, Luna changes the parallel structure of the global attention in Longformer/BigBird to a serial structure. Also, Luna does not pass the updated global tokens ($P'$) through the FF layer as in BigBird; instead, $P'$ is output directly.

For PoNet, instead of introducing additional tokens to preform cross-attention, we use the results of average pooling as the global representation of the first stage of GA. Also, in the second stage of GA, the global representation is served as query in GA rather than key and value in Luna to calculate cross-attention. While Luna is able to directly obtain an output of the same length as the sequence, our GA obtains an output consistent with the length of the first stage global representation (i.e., 1), so we further integrate this output into the original input by element-wise multiplication.

### D.4 DIFFERENCE BETWEEN PONET AND FNET

To the best of our knowledge, our work is the first to comprehensively explore the full potential of the simple pooling mechanism for token mixing and verify that built upon multi-granularity pooling and pooling fusion, our PoNet is competitive in modeling long-range dependency with a linear complexity. The token mixing mechanism in PoNet, built upon pooling mechanisms, is completely different from Fourier transforms used by FNet.

As shown in Table 1, on the LRA benchmark, PoNet achieves an accuracy of 61.05, which is significantly better than the accuracy from FNet (55.30 reported in the original paper with JAX/Flax implementation, 53.10 in our PyTorch implementation). PoNet achieves significantly better pre-training accuracy (with the same MLM+SSO tasks) than FNet, as shown in Figure 2. PoNet achieves significantly better transfer learning performance on GLUE than FNet, as shown in the updated Table 3, and significantly better transfer learning performance on long-text classification tasks than FNet, as shown in Table 4.

## E  ADDITIONAL MODEL CONFIGURATIONS FOR PONET

We explore several additional ideas to improve PoNet.

### E.1 TREE MAX-POOLING (TMP)

For different layers in PoNet, we apply different dilation sliding window max-pooling as another way to exchange the information between two different tokens. We compute the TMP value $T \in$

$\mathbb{R}^{N \times d}$. The size (length) of the dilation windows is based on 2, that is, on the $l$-th layer, $len_{dilation} = 2^{l-1}$. The lowest level, i.e. Level 1, uses a length of 1 for dilation, which is a normal sliding window max-pooling. Since each length of the dilation sliding windows can be represented by binary and the connection state of tokens can be represented as $\{0, 1\}$ (not link or link), the distance between any two tokens can be reached by this structure. We can easily calculate that the longest distance that the structure could reach was $2^{l+1} - 2$. After adding TMP into pooling fusion for PoNet, we observed that the MLM validation accuracy improves in pre-training, but we did not observe performance improvements on the downstream GLUE tasks.

### E.2 CONTEXTUAL DECISION MECHANISM

Since different tokens may need different levels of information, strictly adding the three pooling features together for all tokens, as in Equation 9, cannot meet this requirement. We consider another aggregation method, denoted Contextual Decision Mechanism, to replace the Pooling Fusion in Section 3. Following the idea of attention, each token conducts a cross-attention with the 3 pooling features for contextual interactions, as follows:

$$\boldsymbol{M'_n} = Attention\{\boldsymbol{Q_{Hn}}, \boldsymbol{K_{Mn}}, \boldsymbol{V_{Mn}}\} \in \mathbb{R}^d, \tag{12}$$

where

$$\boldsymbol{Q_H} = \boldsymbol{H}\boldsymbol{W_{K_H}} + \boldsymbol{b_{K_H}}, \tag{13}$$
$$\boldsymbol{K_{Mn}} = [\boldsymbol{g'}; \boldsymbol{S_n}; \boldsymbol{L_n}]\boldsymbol{W_{K_M}} + \boldsymbol{b_{K_M}}, \tag{14}$$
$$\boldsymbol{V_{Mn}} = [\boldsymbol{g'}; \boldsymbol{S_n}; \boldsymbol{L_n}]\boldsymbol{W_{V_M}} + \boldsymbol{b_{V_M}}, \tag{15}$$

and $\boldsymbol{W_*} \in \mathbb{R}^{d \times d}$, $\boldsymbol{b_*} \in \mathbb{R}^d$ are parameters to be learned. Note that $\boldsymbol{K_{Mn}}, \boldsymbol{V_{Mn}} \in \mathbb{R}^{3 \times d}$. $\boldsymbol{M'}$ is then the final output of the multi-granularity pooling block. Similar to the TMP idea, when switching from the pooling fusion in Section 3 to this contextual decision mechanism for PoNet, we observed that the MLM validation accuracy improves in pre-training, but did not observe performance improvements on the downstream GLUE tasks.

## F FUTURE UPDATES FOR PONET

### F.1 REDUCING THE NUMBER OF PARAMETERS

Note that after sharing $\boldsymbol{W_{K_g}}, \boldsymbol{b_{K_g}}$ and $\boldsymbol{W_{V_g}}, \boldsymbol{b_{V_g}}$ in Equation 1, our current PoNet model has a total of 5 linear transformations from Equation 1 in the token mixing layer. Note that other transformations, such as $\boldsymbol{W_{K_g}}$ and $\boldsymbol{W_s}$, are difficult for sharing, because the corresponding projection outputs from $\boldsymbol{W_{K_g}}$ and $\boldsymbol{W_s}$ are used for different procedures of global cross-attention and segment max-pooling, respectively. Sharing them will cause difficulty in parameter updating. For other parameters, since $\boldsymbol{W_{Q_g}}$ will only be applied on a single token (global token) instead of on the whole input sequence, its computational cost can be ignored. The main computational cost comes from the remaining 4 transformations. To further reduce the number of parameters of PoNet, we plan to also remove $\boldsymbol{W_o}$. This is based on the consideration that the original input sequence is not involved in the computations of GA, SMP, and LMP, hence it can be mixed directly with GA and SMP without transformations. This leaves us with only 3 sets of linear transformation parameters that require the computations of the whole sequence, namely, shared $\boldsymbol{W_{K_g}}$ and $\boldsymbol{W_{V_g}}$, $\boldsymbol{W_s}$, and $\boldsymbol{W_l}$. Then PoNet will have the same number of linear transformations as BERT, which has a total of 3 linear transformations in the self-attention layer. We will also investigate sharing these linear transformations across different PoNet layers, which will further reduce the number of parameters. We plan to run more experiments to evaluate the effect on the performance of PoNet from these methods on reducing the number of parameters.

### F.2 SUPPORT CAUSAL ATTENTION

The key issue for supporting causal attention is to prevent information leakage from the moment $t > T$ to the sequence $t <= T$. If the computational complexity of the complete sequence is $O(N)$, to ensure that the newly added $t > T$ moments still maintain the $O(N)$ complexity, the computation

cost must be guaranteed to be $O(1)$ at the introduction of the $T + 1$ moments, i.e., it needs to be possible to recursively push out the results from $t <= T$ to the $T + 1$ moments.

Considering the three different pooling types in PoNet, firstly, since LMP and SMP do not actually introduce multiplication, the computational complexity is negligible. In order to guarantee that the complexity of the Max-pooling operation is $O(N)$, we can make the following improvements. For LMP, it is sufficient to change the pooling window to a causal window without adding extra computation. For SMP, since $\max(S_{t<=T+1}) = \max(S_T, \max(S_{t<=T}))$, the additional token computation can be bounded to $O(1)$ by accumulating $\max$, and the overall computation $O(N)$ will not change. However, for GA, it's not feasible since there are two variables involved in the calculation of attention in cross-attention, one of which is a sequence of length $T$. The T+1 moment inevitably introduces $O(T)$ computation, which makes the overall computational complexity rise to $O(N^2)$. We consider a weaker model scheme by removing SS-GA (PoNet w/o SS-GA) (also mentioned in section 5). As shown in Table 5, PoNet w/o SS-GA has no performance degradation over the original model on MLM and single-sentence classification tasks, i.e. CoLA. These two tasks reflect the ability of sentence encoding, which is important for auto-regressive decoding task. PoNet w/o SS-GA will only have additive calculation in the GA stage, and the computational complexity is negligible. Similar to SMP, we can also simplify the computation by using the recursive property of mean, as $\mathrm{mean}(S_{t<=T+1}) = \frac{T*\mathrm{mean}(S_{t<=T})+S_{T+1}}{T+1}$. Overall, this causal attention in PoNet has $O(N)$ computational complexity.

