# OpenReview forum: "PoNet: Pooling Network for Efficient Token Mixing in Long Sequences"
_ICLR.cc/2022/Conference — ICLR 2022 Poster_

### Official Review · Reviewer_ssBH · 2021-10-31

**Correctness:** 3
**Technical Novelty And Significance:** 2
**Empirical Novelty And Significance:** 3
**Recommendation:** 5
**Confidence:** 4

**Main Review:**

Strengths:
1. Their method is easy to understand and highly motivated.
2. Extensive experiments have been conducted and reasonable performance has been obtained.
3. Ablation studies show the contribution of these pooling mechanisms.

Weakness:
1. PoNet Is suitable for language representation learning but to my understanding, it is impossible for PoNet to build generation models such as machine translation.
2. I do not find the number of parameters for PoNet. Will these pooling methods introduce a lot of extra parameters?
3. Some relevant works should be discussed, such as Luna: Linear Unified Nested Attention.
4. More importantly, the authors introduce the SSO during the pertaining. Is the FNet pertaining also using this objective? If not, I think the comparison is not fair.

Minor comments:
In the abstract, authors claim that "PoNet significantly outperforms Transformer" which should be in terms of memory and speed, right? Maybe here could be clearer.

**Summary Of The Paper:**

In this paper, the authors aim to resolve the quadratic time and memory complexity of the standard attention mechanism. They introduce a multi-granularity pooling and pooling fusion that captures contextual information from various levels (global, segment, and token).  They conduct experiments on long sequence tasks and large-scale pre-training and fine-tuning. As a result, their model achieves great performance with speed-up.

**Summary Of The Review:**

The authors introduce a pooling-based module to replace the self-attention layer in the Transformer in order for fast training speed and low memory usage. Extensive experiments show the effectiveness of their methods. However, some details of their models should be further clarified in the paper (such as the number of parameters).

---

> ### Author Response · Authors · 2021-11-20
> **Response to Reviewer ssBH (1/3)**
>
> We sincerely thank the Reviewer for all the comments. We address all the points below:
>
> ### Re Weakness 1
>
> *"PoNet Is suitable for language representation learning but to my understanding, it is impossible for PoNet to build generation models such as machine translation.”*
>
> In this paper, we focus on developing a novel, effective and efficient drop-in replacement for self-attention in Transformer encoder.  For generation tasks, we plan to first evaluate using PoNet for encoder and the standard full self-attention Transformer decoder in the encoder-decoder framework on summarization tasks, similar to how Longformer and BigBird were evaluated on summarization tasks.  To improve efficiency such as for machine translation, we plan to investigate replacing self-attention in Transformer encoder with PoNet, and using PoNet for auto-regressive attention, **described in Appendix F.2 on how to support auto-regressive attention in PoNet**,  for decoder,  as well as replacing encoder-decoder cross-attention with a pooling mechanism.
>
> ### Re Weakness 2
>
> *“I do not find the number of parameters for PoNet. Will these pooling methods introduce a lot of extra parameters?”*
>
> Thank you for the question. The number of parameters for PoNet-Base is 124M, comparable to the number of parameters for BERT-Base as 110M. We updated Section 4 to include this information. As shown in Section 3.1, multi-granularity pooling brings six groups of parameters ($W,b$) in Eq.1 to be learned, while self-attention in BERT only brings three groups. As shown in the first paragraph of Appendix A, we share the parameters of {$W_{K_g}, b_{K_g}$} and {$W_{V_g}, b_{V_g}$}  in Eq. 1 for all experiments on PoNet in the paper and we observe no performance degradation.  After this sharing of {$W_{K_g}, b_{K_g}$} and {$W_{V_g}, b_{V_g}$}  in Eq. 1, our current PoNet model has a total of 5 linear transformations from Eq.1 in the token mixing layer, and has 124M model parameters.
>
> We describe how to further reduce the number of parameters in Appendix F.1.  Note that other transformations, such as $W_{K_g}$ and $W_s$, are difficult for sharing, because the corresponding projection outputs from $W_{K_g}$ and $W_s$ are used for different procedures of global cross-attention and segment max-pooling, respectively. Sharing them will cause difficulty in parameter updating. For other parameters, since $W_{Q_g}$ is applied on a single token (global token) instead of on the whole input sequence, its computational cost can be ignored. The main computational cost comes from the remaining 4 transformations. To further reduce the number of parameters of PoNet, we plan to also remove $W_o$. This is based on the consideration that the original input sequence is not involved in the computations of GA, SMP, and LMP,  hence it can be mixed directly with GA and SMP without transformations. This leaves us with only 3 sets of linear transformation parameters that require the computations of the whole sequence, namely, shared $W_{K_g}$ and $W_{V_g}$ , $W_s$ , and $W_l$ . Then PoNet will have the same number of linear transformations as BERT, which has a total of 3 linear transformations in the self-attention layer. We will also investigate sharing these linear transformations across different PoNet layers, which will further reduce the number of parameters.  We will evaluate the effect on the performance of PoNet from these methods on reducing the number of parameters.

---

> ### Author Response · Authors · 2021-11-29
> **Response to Reviewer ssBH (2/3)**
>
> ### Re Weakness 3
>
> *“Some relevant works should be discussed, such as Luna: Linear Unified Nested Attention.”*
>
> Thank you for your suggestion. We have added this citation into Section 2 related work, and also added discussions on algorithm comparisons between PoNet and Luna to Appendix D.3, and compared PoNet with Luna on the LRA results on Page 7 in text,  in the updated draft.
>
> **However, we want to emphasize the cautions when comparing the results on LRA and GLUE between our PoNet and Luna.**
>
> **LRA results**
>
> Note that in our paper,  all results in the third group in Table 1, including baseline models and our PoNet, are fairly compared, since all models in this group are implemented based on the same PyTorch codebase based on (Xiong et al., 2021) and evaluated using the same configurations as used by (Xiong et al., 2021).
>
> **However, these results in the third group of Table 1 cannot be fairly compared to the LRA results from Luna (Ma et al., 2021), due to the following reasons.** Firstly, we use Xiong et al. 's PyTorch codebase whereas Luna follows Tay et al.’s implementations (in JAX/Flax), which are not directly comparable. Hence, it is more reasonable to compare the gain over the respective vanilla Transformer results. Secondly, Luna did an adjustment to the LRA evaluation and obtained much better results ($59.24$) from their re-implemented Transformer over the original Transformer ($54.39$), as explained below in the Luna paper:
>
> >cite Luna: “For the task of Retrieval, we ﬁnd that models are not fully converged when being trained for 5K steps as stated in Tay et al. (2021). Therefore, we train models for 20K steps for this task and obtain much better results.”
>
> Since the authors apply this adjustment to Transformer (re-impl) and all Luna-L evaluations, it is reasonable to compare our PoNet’s gain over our Transformer baseline to Luna’s gain over their Transformer baseline. As shown in the third group in Table 1, our PoNet achieves $\boldsymbol{+2.28}$ gain ($61.05$ over $58.77$) on the average accuracy over the vanilla Transformer.  In the Luna paper, their reimplemented Transformer achieves $59.24$ AVG,  hence Luna-16 ($61.46$ AVG), Luna-128 ($61.93$), Luna-256 ($61.95$) achieve $+2.22$, $+2.69$, and $+2.71$ gain over their re-implemented Transformer, respectively. Hence, the gain from PoNet over Transformer on LRA is quite close to the gain Luna achieves over Transformer.
>
> **Comparing the training speed and peak memory consumption between PoNet and Luna**, based on comparing Table 2 in our paper and Table 2 in the Luna paper,  it can be observed that PoNet is significantly faster than Luna. With larger projected dimensions, i.e. 128 and 256, Luna requires more memory. When comparing PoNet with Luna-128 and Luna-256, PoNet is heavier than Luna for $1K-2K$ input sequence length but already lighter than Luna for $>2K$ input sequence length. PoNet is significantly faster than Luna, with PoNet reaching $\boldsymbol{4.4\times}$ for $2K$ and $\boldsymbol{9.0\times}$ for $4K$ input sequence length, whereas Luna-128 is $3.4\times$ for $2K$ and $5.1\times$ for $4K$ input sequence length.
>
> **GLUE results**
>
> We also want to mention that regarding the GLUE results, the maximum sequence length of the GLUE tasks is around 128, while Luna uses 128 additional tokens as global tokens for GLUE evaluations (Luna paper, Section 4.3), hence the amount of computations of Q, K, V in Luna-128 are already consistent with that of Transformer, on top of the fact that Luna does cross-attention twice in the token-mixing layer.

---

> ### Author Response · Authors · 2021-11-29
> **Response to Reviewer ssBH (3/3)**
>
> ### Re Weakness 4
>
> *“More importantly, the authors introduce the SSO during the pertaining. Is the FNet pertaining also using this objective? If not, I think the comparison is not fair.”*
>
> Yes, in Figure 2, FNet also uses the same pretraining task (MLM+SSO), the same pre-training data Wikitext-103(100M words)+BooksCorpus (800M words), in total 5GB data size,  and the same pre-training configurations (Appendix A.2), as for BERT and PoNet for a fair comparison. Figure 2 shows that FNet is much worse than PoNet in both MLM and SSO accuracies. In the updated Table 3, same as Figure 2, BERT-Base, FNet-Base, and PoNet-Base are all pre-trained with the same **5GB** pre-training data Wikitext-103(100M words)+BooksCorpus(800M words), the same pre-training tasks of MLM+SSO, and the same pre-training configurations as used for Figure 2 (Appendix A.2), all pre-trained with **340K steps**.  All three models are base model uncased.  We run 20 sets of hyper-parameter configurations based on Table 6 for fine-tuning BERT, FNet, and our PoNet and report the best GLUE results for each of them in Table 3. The comparison is fair, and it shows that the AVG score for BERT is $80.21$, $76.80$ for PoNet, and $73.46$ for FNet. We observe that PoNet achieves **95.7%** of the accuracy of BERT on the GLUE benchmark, whereas FNet achieves only 91.6% of the accuracy of BERT on GLUE and PoNet outperforms FNet by **4.5%** relatively.
>
> Note that due to limited machine resources available to us and for achieving faster turnaround for the extensive experiments and ablation studies we conducted, we used the small 5GB data for pre-training PoNet.  The reason that the $80.21$ AVG from BERT in Table 3 is about 2 points lower than the number reported either by the BERT paper or by Huggingface  is due to the fact that the BERT model in Table 3  is pre-trained with much smaller data 5GB data than the 16GB data as used by standard BERT, and pre-trained with much fewer steps, 340K steps, than the 1M steps as by standard BERT.
>
>
> **As to the concern that the capacity of Transformer requires a large amount of pre-training data to demonstrate, this is the purpose of the second group with (2) in Table 7, shown on Page 16 in Appendix B.** In the second group with (2) in Table 7, the GLUE fine-tuning results for BERT and FNet are from fine-tuning the official checkpoints of BERT-Base and FNet-Base, respectively (see Appendix B for details). It is important to point out that the GLUE results from official checkpoints of BERT-Base and FNet-Base and our pre-trained PoNet-Base are not comparable, since the pre-training data and pre-training tasks are different. BERT-Base was pre-trained on English Wikipedia (2.5B words) +BooksCorpus (800M words) with MLM+NSP, in total 16GB data size. FNet-Base was pre-trained on C4 (~700GB data size) with MLM+NSP. Hence, **the capacity of both BERT and FNet models in the second group of Table 7 is considered to be fairly fully demonstrated.**  In contrast, our PoNet-Base was pre-trained on 5GB Wikitext-103+BooksCorpus with MLM+SSO. Nevertheless, with the much smaller pre-training data (5GB data), PoNet-Base still achieves GLUE AVG $77.54$, reaching **92.84%** of the accuracy of BERT (16GB data) on GLUE (83.52) and outperforms FNet (74.23) by **4.5%** relatively, and also better than 76.7 reported in the original FNet paper (they used different code implemented with JAX/Flax). These results prove the promising potential of PoNet.
>
> More details on fine-tuning the official checkpoints can be found in Appendix B. Also, in Appendix B, we verified the correctness of our fine-tuning results for BERT and FNet official checkpoints by comparing our results to the results reported by HuggingFace.  As shown in the second group of Table 7,  for the official BERT-Base-uncased checkpoints, we obtain the best GLUE validation results as AVG **83.52**, better than 82.62 HuggingFace reported on GLUE validation set from BERT-Base-cased (https://huggingface.co/google/fnet-base).
>
> We are in the process of further investigating the capacity of PoNet and pre-training PoNet with exactly the same pre-training data (16GB data) as used by BERT. It would be definitely useful to finish this experiment and we will report GLUE fine-tuning results once this experiment is completed.
>
> ### Re Minor comments
>
> *"In the abstract, authors claim that "PoNet significantly outperforms Transformer" which should be in terms of memory and speed, right? Maybe here could be clearer.”*
>
> Thank you for your comments. Yes, PoNet significantly outperforms Transformer in accuracy in modeling long-range dependencies (see Table 1), and also in memory and speed (see Table 2). We have updated the paper.
>
> Thanks!

---

### Official Review · Reviewer_vM28 · 2021-11-02

**Correctness:** 3
**Technical Novelty And Significance:** 2
**Empirical Novelty And Significance:** 3
**Recommendation:** 6
**Confidence:** 4

**Details Of Ethics Concerns:**

None.

**Main Review:**

Strengths:

1. This paper accelerates calculation and retains accuracy in common benchmarks.
2. The paper is clear.
3. Although the idea of each module in multi-granularity pooling is not entirely new, the combination is novel and comprehensive.

Weaknesses:

1. The improvements are not significant over the previous token-mixing method, the FNet. Moreover, their experimental settings are different, and the FNet gets lower performance in this setting. As to the originality, FNet does token-mixing, the contribution of Ponet is incremental.
2.  The paper relies on empirical results but does not report error bars.
3. The paper claims it designs a new pretraining objective in their abstract, but they do not validate this claim well.
4. They do not report their speedup on the GLUE benchmark (e.g., reduces xx% GPU/TPU time) but only report that they lose 4% accuracy than the Transformer baseline.
5. This paper lacks theoretical evidence or understandings.

Minor:
The right subfigure of Figure 1 is a bit confusing.


**Summary Of The Paper:**

This paper proposes PoNet, an efficient model with linear complexity for modeling long sequences. This model replaces the self-attention layer of the Transformer model with its pooling network.

To aggregate information from different levels, their pooling network consists of three pooling strategies:
a. Global aggregation that aggregates information from all tokens in a sequence. This method is a bit like BERT CLS, and the difference is that they use average representations rather than the representation of the first token.
b. Segment Max-pooling that performs max-pooling in each sentence to capture sentence-level information.
c. Local Max-pooling that performs max-pooling in each window to capture local information.
Their fusion method is the addition operation.

As to the training objective, they not only use the MLM, but also use the sentence structural objective as in StructBERT.

This paper contributes to a well-defined but highly-influential problem: Efficient modeling for long sequences. The main contribution of this paper is the introduction of a multi-granularity pooling that is more efficient than self-attention and brings higher accuracy than previous efficient methods.

Empirical studies are performed to show the superiority of PoNet over previous SOTA FNet. Their method outperforms FNet in terms of accuracy in their experiment settings but slightly lags regarding time efficiency.
Results are shown on the task of (1) Long-range Arena (2) GLUE by fine-tuning pre-trained networks.


**Summary Of The Review:**

The paper is good but not enough. It empirically improves the accuracy of classification and language modeling. But the significance and the originality are not high, and they do not bring us much new knowledge. Their comparisons to baselines are not sufficient, e.g. they can be benefited from reporting GPU/TPU time. Since it does not meet the high standard of ICLR, I tend to reject this paper.

---------------Post Rebuttal-------------

The authors have clarified my concerns about performance, I decide to raise the score from 5 to 6.  I still recommend authors to provide code, report error bars, speedup on GLUE. Moreover, the combination of StructBERT and MLM can not be a significant contribution. Finally, the challenge of combining these pooling methods and pertaining objectives should be discussed and well-evidenced.

---

> ### Author Response · Authors · 2021-11-20
> **Response to Reviewer vM28**
>
> We sincerely thank the Reviewer for all the comments. We address all points raised in weaknesses below.
>
> ### Re Weakness 1
>
> *"The improvements are not significant over the previous token-mixing method, the FNet. Moreover, their experimental settings are different, and the FNet gets lower performance in this setting. As to the originality, FNet does token-mixing, the contribution of Ponet is incremental."*
>
> **Significant Improvements over FNet:**
>
> As shown in Table 1, on the LRA benchmark, PoNet achieves an accuracy of 61.05, significantly better than the accuracy from FNet (55.30 reported in their paper with JAX/Flax implementation, 53.10 from our PyTorch implementation). PoNet achieves significantly better pre-training accuracy (with the same MLM+SSO tasks) than FNet, as shown in Figure 2. PoNet achieves significantly better transfer learning performance on GLUE than FNet, as shown in the updated Table 3, and significantly better transfer learning performance on long-text classification tasks than FNet, as shown in Table 4.
>
> **Originality of PoNet:**
>
> First, to the best of our knowledge, our work is the first to comprehensively explore the full potential of the simple pooling mechanism for token mixing and verify that built upon multi-granularity pooling and pooling fusion, our PoNet is competitive in modeling long-range dependency with a linear complexity. The token mixing mechanism in PoNet, built upon pooling mechanisms, is completely different from Fourier transforms used by FNet.
>
> Second, our work aims at achieving two goals simultaneously. The first goal is to better balance the tradeoff between model quality and efficiency, achieving advantages over Transformer and other efficient transformers in accuracy in long-range modeling tasks and memory consumption and speed. The second goal is to achieve competitive transfer learning capabilities, which are crucial for NLP performances. We verify that PoNet is able to achieve both goals, demonstrating competitive transfer learning capabilities and producing competitive performance on both downstream single-sentence and sentence-pair tasks.
>
> ### Re Weakness 2
>
> *"The paper relies on empirical results but does not report error bars."*
>
> We follow the conventions in previous efficient transformer works and did not report error bars.
>
> ### Re Weakness 3
>
> *"The paper claims it designs a new pretraining objective in their abstract, but they do not validate this claim well."*
>
> First, to clarify, we use MLM and sentence structural objective (the same in StructBERT) for pre-training PoNet. These pre-training objectives are not new. The novelty is to understand the interactions between model architecture and pre-training tasks and decide pre-training tasks. For the over-redundant full self-attention mechanism in BERT, SSO may not contribute much on top of MLM (StructBERT Table 4). However, the pooling mechanism in PoNet is simple hence we hypothesize that it is crucial to couple MLM with SSO to balance a good tradeoff between sing-sentence and sentence-pair tasks.
>
> We conduct ablation study on efficacy of MLM and SSO tasks. The last two rows of Table 5 show that the weakened SST loss (MLM+NSP or Only MLM) weakens GA representation learning while strengthening SMP and LMP learning, causing a significant degradation in STS-B accuracy but a significant gain in CoLA accurcy. Overall on single-sentence tasks and sentence-pair tasks, PoNet pre-trained with MLM+SSO significantly outperforms PoNet pre-trained with MLM+NSP or only MLM, showing the importance of pre-training PoNet with MLM+SSO.
>
> ### Re Weakness 4
>
> *“They do not report their speedup on the GLUE benchmark (e.g., reduces xx% GPU/TPU time) but only report that they lose 4% accuracy than the Transformer baseline. “*
>
> We reported in detail the speed comparisons for different sequence lengths on the LRA text classification task, so it may be redundant to report speedup on the GLUE benchmark again.
>
> Table 2 compares the GPU training speed and peak memory consumption of PoNet to Transformer, Performer, Nystromformer, and FNet on a single NVIDIA V100 chip, on input sequence lengths from 512 up to 16384. We observe that PoNet is the second fastest model and consumes the second smallest memory footprint in the group, consistently on all sequence lengths, much faster than Transformer, Performer, and Nystromformer and lighter than them, and only slightly slower and heavier than FNet. Also, the speedup from PoNet over Transformer escalates on longer input sequence lengths, with efficiency up to 9 times and memory consumption 10% of that of Transformer on GPU with 4K input sequence length, as shown in Table 2.

---

> > ### Author Response · Authors · 2021-11-23
> > **Continue Response to Reviewer vM28**
> >
> > ### Re Weakness 5
> > *“This paper lacks theoretical evidence or understandings.”*
> >
> > We show the theoretical complexity analysis in Section 3.2.  The first paragraph in Section 3 describes the motivation for PoNet. Our work is inspired by External Attention proposed in (Guo et al, 2021) for visual tasks. We find that softmax can effectively model token-mixing. Since computation of softmax is still slow, we replace softmax with a simpler pooling mechanism. Furthermore, our design of the multi-granularity pooling and pooling fusion is inspired by previous works such as Longformer and BigBird.  Our design considers the importance of capturing context information from different granularities and combining the captured information.  Comprehensive evaluations and ablation studies verify the designed multi-granularity pooling and pooling fusion is necessary for achieving good performance on both downstream single-sentence and sentence-pair tasks in transfer learning.
> >
> > ### Re Minor
> > *“Minor: The right subfigure of Figure 1 is a bit confusing.”*
> >
> > Thank you for your comment. We will try to update Figure 1 to make it clearer.
> >
> > Thanks!

---

> ### Comment · Reviewer_vM28 · 2021-11-29
> **Questions about the significantly low performance of BERT-base  baseline on GLUE**
>
> Questions: I'm confused as to why your results for BERT-base on GLUE  in Table 3  are significantly weaker than the results reported either by the BERT paper or by the implementation of Huggingface. On average, there is a gap of 2 points.

---

> > ### Author Response · Authors · 2021-11-29
> > **Re: Reviewer vM28**
> >
> > Thank you for your question!
> >
> > 1. Table 3 shows that the AVG score for BERT is $80.21$, $76.80$ for PoNet, and $73.46$ for FNet. **The reason that this $80.21$ AVG from BERT is about $2$ points lower than the number reported** **either by the BERT paper or by the implementation of Huggingface**  **is due to the fact that the BERT model in Table 3 is pre-trained with much smaller data $5GB$ data than the $16GB$ data as used by the standard BERT model, and pre-trained with much fewer steps, $340K$ steps, than the $1M$ steps as by the standard BERT model.**
> >    Due to limited machine resources available to us and for achieving faster turnaround for the extensive experiments and ablation studies we conducted, we used the small Wikitext-103 ($100M$ words) and Bookscorpus ($800M$ words), in total $\sim 5GB$ data size, for pre-training PoNet. As stated in the discussions on Table 3 (Section 4.2, page 7 and 8) and in our rebuttal, the purpose of Table 3 and Figure 2 is to facilitate a fair comparison between BERT, FNet, and PoNet. All three models BERT, FNet, and PoNet in Table 3 are pre-trained using the same $5GB$ pre-training data (Wikitext-103 and Bookscorpus), using the same pre-training tasks MLM+SSO, and pre-trained with $340K$ steps with the same hyper-parameters (see Appendix A.2 for details). And the GLUE results are from fine-tuning these pre-trained models with the same configurations (see Appendix A.3 for details). Hence, the GLUE results in Table 3 are comparable and we observe PoNet achieves $95.7$% of the accuracy of BERT on GLUE, whereas FNet achieves only $91.6$% of the accuracy of BERT on GLUE.
> >
> > 2. **We also reported GLUE results from fine-tuning the official checkpoints of BERT and FNet in the second group with (2) in Table 7 in Appendix B**, as summarized in the paper updates on the portal page. Note that the results with (2) in the second group of Table 7 is from fine-tuning the official Huggingface BERT-Base-uncased checkpoints (https://huggingface.co/bert-base-uncased) ($16GB$ data, $1M$ steps), the official Google FNet-Base-cased checkpoints ([https://github.com/google-research/google-research/tree/master/f_net](https://github.com/google-research/google-research/tree/master/f\_net)) ($700GB$ data, $1M$ steps), and our pre-trained PoNet ($5GB$ data, $750K$ steps), hence the results are not directly comparable. PoNet-Base still achieves GLUE AVG $77.54$, reaching $92.84$% of the accuracy of BERT on GLUE ($83.52$). Note that for the official BERT-Base-uncased checkpoints, we obtain the best GLUE validation results as AVG **$83.52$**, which is better than the $82.62$ AVG score HuggingFace reported on GLUE validation set from BERT-Base-cased (https://huggingface.co/google/fnet-base). These results verified the correctness of our BERT fine-tuning implementations. More details on fine-tuning the official checkpoints can be found in Appendix B. Also, in Appendix B, we verified the correctness of our fine-tuning results for BERT and FNet official checkpoints by comparing our results to the results reported by HuggingFace.
> > 3. Since the second group of Table 7 reports fine-tuning the official checkpoints of BERT and FNet, the capacity of both models is considered to be relatively fully demonstrated. Nevertheless, with the much smaller $5GB$ pre-training data and fewer training steps of $750K$, PoNet-Base still achieves GLUE AVG $77.54$, reaching **$92.84$%** of the accuracy of BERT on GLUE ($83.52$) and outperforms FNet ($74.23$) by $4.5$% relatively, and also better than $76.7$ reported in the original FNet paper (they used different code implemented with JAX/Flax). These results prove the promising potential of PoNet. We are in the process of pre-training PoNet with the same $16GB$ pre-training data as BERT, and will update the GLUE results when they are ready.
> >
> > Thanks!

---

> > > ### Comment · Reviewer_vM28 · 2021-11-29
> > > **Thank you for clarifying conerns about improvemnts.**
> > >
> > > I decide to raise my score from weak reject to weak accept.

---

### Official Review · Reviewer_Xvc7 · 2021-11-03

**Correctness:** 3
**Technical Novelty And Significance:** 2
**Empirical Novelty And Significance:** 2
**Recommendation:** 5
**Confidence:** 5

**Main Review:**

The architecture of PoNet is simple and the paper is clearly written.

However, there are some weaknesses:

1. The proposed architecture of PoNet is not novel. The key component, which is the global aggregation module, is very similar to FastFormer. The other two components are also straight-forward and less important than GA.

2. The experimental results are not entirely convincing, especially those on large-scale pre-training and fine-tuning. The BERT-base model in the second group of Table 3 is trained on the same data of the original BERT paper, with and advanced SSO loss and more update steps (125k in original BERT paper vs. 340k in this paper). However, the fine-tuning performance of the BERT-base in Table 3 on GLUE is even worse than that in the original paper. In the paper of Linformer (Wang et al., 2020, Table 2), they re-trained a BERT-base model with only the MLM loss and 250k update steps, but their scores on GLUE are also better than that in Table 3.

3. The authors did not clarify if the PoNet can be used in auto-regressive attention.

4. Personally speaking, some claims in this paper are not well-supported. For example, the last sentence in page 2 claims that the low-rank approximations used in Linformer and Nystromformer limit the ability of looking at the full sequence and hence degrade long-range modeling capabilities. I do not see why PoNet is better than these low-rank methods on modeling long-range modeling.

Questions:

1. In the experiments of LRA, in Table 1 the authors said that they used the same setting as (Xiong et al., 2021). However, in Table 2 they said the setting is the same as (Tay et al., 2021). Are these two experiments using different settings?

2. In table 2, Performer is 4.3 times faster than Transformer under 4K sequence length. However, in the original LRA paper (Tay et al., 2021), the number is 5.7. What is the difference?

3. In the LRA paper, the authors clarified that the sequences in the text classification task are trunked to length 4K. Why in Table 2, there are two columns with 8K and 16K lengths?

Missing references:

There are some relevant papers that are missed in the related work.

Zhu et al., Long-Short Transformer: Efficient Transformers for Language and Vision. NeurIPS 2021.

Ma et al., Luna: Linear Unified Nested Attention. NeurIPS 2021.


**Summary Of The Paper:**

This paper proposed PoNet, which is an efficient architecture to replace self-attention in Transformer-based models.
PoNet consists of three components, which are called multi-granularity pooling block.

The first pooling component is the global aggregation module, which is very similar to the additive attention in FastFormer. The different is that in FastFormer, the global query vector is computed by a weighted sum of the query vectors, while in PoNet it is the average of the query vectors. The segment max-pooling and local max-pooling modules are straight-forward, which are max-pooling operations on pre-defined segments and on local sliding windows, respectively.

Experiments were conducted on Long-range Arena (LRA) and large-scale pretraining and fine-tuning.
The authors also conducted ablation experiments to analyze the contributions of these three multi-granularity pooling components.

**Summary Of The Review:**

To sum up, the idea in this paper is incremental and more empirical evidences are needed to demonstrate the effectiveness of the proposed PoNet architecture.

---

> ### Author Response · Authors · 2021-11-20
> **Response to Reviewer Xvc7 (1/3)**
>
> We sincerely thank the Reviewer for all the comments. Below we address all points raised in weaknesses and questions.
>
> ### Re Weakness 1
>
> *"The proposed architecture of PoNet is not novel. The key component, which is the global aggregation module, is very similar to FastFormer. The other two components are also straight-forward and less important than GA."*
>
> **The Novelty of PoNet:**
>
> First, to the best of our knowledge, our work is the first to comprehensively explore the full potential of the simple pooling mechanism for token mixing and verify that built upon multi-granularity pooling and pooling fusion, our PoNet is competitive in modeling long-range dependency with a linear complexity. The token mixing mechanism in PoNet, built upon pooling mechanisms, is completely different from Fourier transforms used by FNet.
>
> Second, our work aims at achieving two goals simultaneously. The first goal is to better balance the tradeoff between model quality and efficiency, achieving advantages over Transformer and other efficient transformers both in accuracy in long-range modeling tasks and memory consumption and speed. The second goal is to achieve competitive transfer learning capabilities, which are crucial for NLP performances. We verify that PoNet is able to achieve both goals, demonstrating competitive transfer learning capabilities and producing competitive performance on both downstream single-sentence and sentence-pair tasks.
>
> **Difference between GA and Fastformer:**
>
> First, our PoNet consists of three components, i.e., global aggregation(GA), segment max-pooling(SMP), and local max-pooling(LMP). We will clarify the importance of SMP and LMP later. Second, the GA component in PoNet is different from Fastformer due to different motivations and implementations, as follows.
>
> In the motivation aspect, our GA is inspired by the global attention in Longformer and BigBird, but is also different from them (see Appendix D.2). In contrast,  Fastformer uses the additive attention mechanism to model global contexts.
>
> In the aspect of implementation, Firstly, as you said, comparing GA in PoNet with Fastformer, the global query is computed differently. In Fastformer, the global query vector is a weighted sum of all query vectors; whereas, in PoNet, it is an average pooling of all query vectors.
>
> Secondly, the global query is mixed with key and value in different ways.
>
> 1. In Fastformer, element-wise product is performed between the global query vector $q$ and each key vector $k_i$ to obtain a set of vectors $p_i$. A set of learnable vectors are introduced for additive attention to compute weights for $p_i$. After softmax normalization of the weights, the sum of weighted $p_i$ is the global key (GK). Then, element-wise product is performed between GK and each value vector $v_i$ to obtain a set of vectors $U$. After that, $U$ passes through the linear transformation layer to obtain its hidden representations $R$. And the hidden representations $R$ are element-wise summed with $Q$ to obtain the final output of Fastformer.
> 2. In our GA, as described in Section 3.1.1, the first stage value for GA, denoted by $g$ in the paper, serves as a query token to interact with the input sequence through cross-attention. There are several differences from Fastformer. First, standard scaled dot-production is performed on $Q$ and $K$ matrices, and the weight of the scalar is obtained directly. Then, softmax is applied on the scaled dot-product results as weights for $V$. The weighted sum of $V$ is computed as the global representation, denoted by $g'$ in the paper, and there is no process of global key (GK). Finally, as described in Section 3.1.4, we perform element-wise product between $g'$ and $H_{on}$ to compute the final global representation for pooling fusion, where $H_{on}$ is from projecting input embedding of each token through another linear transformation specific for pooling fusion. Different from Fastformer, there is no linear transformation layer nor summation with $Q$.
>
> The above comparisons show that the GA in our PoNet is quite different from Fastformer. In addition to algorithm comparisons, we also experimented with replacing the GA module in our PoNet with Fastformer, and observed that after the same pre-training and fine-tuning configurations, replacing GA with Fastformer has lower performance compared to our PoNet: On GLUE CoLA task, the performance dropped from 45.36 to 43.28; on GLUE STS-B task, the performance dropped from 84.57 to 84.46.

---

> ### Author Response · Authors · 2021-11-20
> **Response to Reviewer Xvc7 (2/3)**
>
> ### Continue Re Weakness 1
>
> **The Importance of Other Two Components SMP and LMP:**
>
> The importance of each granularity can be seen in Table 5, where accuracies of pre-training tasks and accuracies on the downstream single-sentence and sentence-pair tasks are shown. Table 5 shows that removing either SMP or LMP drastically degrades the MLM accuracy during pre-training and single-sentence task accuracy after fine-tuning, while yielding comparable SST accuracy and some accuracy gain on the sentence-pair task STS-B. These results confirm the importance of SMP and LMP for capturing local context information, and SMP and LMP are important for the modeling capabilities of PoNet.
>
> We updated Table 5 to show the accuracies from PoNet with GA only (i.e., both w/o SMP and w/o LMP) for pre-training MLM and SST (as SSO) tasks and downstream single-sentence CoLA and sentence-pair STS-B tasks. It shows that removing both SMP and LMP drastically degrades MLM accuracy (from 59.44 to 43.61) and the single-sentence CoLA task (from 45.36 to 11.36).
>
> **Differences between PoNet and Longformer/BigBird**:
>
> Our design of the multi-granularity pooling and pooling fusion is inspired by previous works such as Longformer and BigBird. Our design considers the importance of capturing context information from different granularities and combining the captured information. However, our design is different from previous works.
>
> Longformer uses a window-based self-attention to capture local context and a task-specific global attention to encode inductive bias about the task. For example, for classification tasks the global attention attends to the [CLS] token and for question-answering tasks, the global attention attends to all question tokens. In contrast, GA in PoNet is task-agnostic.
>
> BigBird combines global, windowed local, and random attentions. BigBird chooses a subset from existing tokens as global tokens or adds additional global tokens such as [CLS]. These global tokens attend to all existing tokens. The GA in PoNet is computed differently from BigBird (see Section 3.1.1 and Section 3.1.4). Also, PoNet uses GA, segment max-pooling and local max-pooling. We experimented with replacing our windowed local max-pooling with random pooling in an early PoNet structure with a small number of pre-training steps, and observed that this change weakened PoNet’s capability of capturing local context and significantly degraded the accuracy of downstream single-sentence CoLA task (from 42.57 to 31.44) while causing a slight gain on the sentence-pair STS-B task (from 80.29 to 80.48). Hence, we did not use random attention in the final structure of PoNet.
>
> ### Re Weakness 2
>
> *"The experimental results are not entirely convincing, especially those on large-scale pre-training and fine-tuning. The BERT-base model in the second group of Table 3 is trained on the same data of the original BERT paper, with and advanced SSO loss and more update steps (125k in original BERT paper vs. 340k in this paper). However, the fine-tuning performance of the BERT-base in Table 3 on GLUE is even worse than that in the original paper. In the paper of Linformer (Wang et al., 2020, Table 2), they re-trained a BERT-base model with only the MLM loss and 250k update steps, but their scores on GLUE are also better than that in Table 3."*
>
> Thank you for the good question. We should have described the original Table 3 (now Table 7 in Appendix) more clearly. The second group of the old Table 3 (now Table 7) shows the results from our experiments of fine-tuning each pre-trained checkpoints provided by authors on GLUE. As described in Appendix B, for the "BERT-Base" row in Table 7, we fine-tuned the HuggingFace BERT-Base-uncased checkpoints (https://huggingface.co/bert-base-uncased). For the "FNet-Base" row in Table 7, we fine-tuned the official FNet-Base checkpoints after converting the official FNet checkpoints using the tool from https://github.com/erksch/fnet-pytorch to be loadable by the PyTorch codebase for fine-tuning. For the "PoNet-Base" row in Table 7, we fine-tuned our pre-trained PoNet on GLUE (based on pre-training details in Appendix A.2 and pre-trained with 750K steps). For FNet-Base and PoNet-Base, we run 20 sets of hyper-parameter configurations based on Table 6 and report the best GLUE results in Table 3 (now Table 7); for BERT-Base, we originally used the default hyper-parameters suggested by HuggineFace code. Then we also run 20 sets of hyper-parameter configurations based on Table 6 on BERT-Base official checkpoints, and we obtain the best GLUE results as AVG 83.52 for BERT-Base, as shown in Table 7.

---

> ### Author Response · Authors · 2021-11-20
> **Response to Reviewer Xvc7 (3/3)**
>
> ### Continue Re Weakness 2
>
> It is important to point out that these GLUE results from official checkpoints of BERT-Base and FNet-Base and our pre-trained PoNet are not comparable since the pre-training data and pre-training tasks are different. BERT-Base was pre-trained on English Wikipedia (2.5B words) +BooksCorpus (800M words) with MLM+NSP, in total about 20GB data size. FNet-Base was pre-trained on C4 (about 700GB data size) with MLM+NSP, whereas our PoNet was trained on Wikitext-103 (100M words)+Bookscorpus (800M words) with MLM+SSO, in total about 5GB data size. Nevertheless, with the much smaller pre-training data, PoNet-Base still achieves GLUE AVG 77.54, better than 74.23 from our PyTorch-based FNet-Base fine-tuning, and better than 76.7 reported in the original FNet paper (they used different code implemented with JAX/Flax).
> Note that we updated Appendix B to show the verification of correctness of our PyTorch fine-tune implementations for BERT and FNet.
>
> In order to achieve a fair comparison of pre-training and GLUE fine-tuning, we updated Table 3 with fine-tuning results from BERT, FNet, PoNet on GLUE validation set, all base-uncased model pre-trained with the same configurations, i.e., same Wikitext-103+Bookscorpus (~5GB data), same MLM+SSO tasks, same pre-training hyper-parameters (Appendix A.2) and all pre-trained with 340K steps. We run 20 sets of hyper-parameter configurations based on Table 6  for fine-tuning BERT, FNet, and our PoNet and report the best GLUE results for each of them in Table 3. The AVG score for BERT is 80.21, 76.80 for PoNet, and 73.46 for FNet. We observe that PoNet achieves 95.7% of the accuracy of BERT on the GLUE benchmark, whereas FNet achieves only 91.6% of the accuracy of BERT on GLUE.
>
> ### Re Weakness 3
>
> *"The authors did not clarify if the PoNet can be used in auto-regressive attention."*
>
> In this paper, we focus on replacing the full self-attention in Transformer encoder with PoNet.  We added Appendix F.2 to describe how to use PoNet in auto-regressive attention in future work.
>
> ### Re Weakness 4
>
> *"Personally speaking, some claims in this paper are not well-supported. For example, the last sentence in page 2 claims that the low-rank approximations used in Linformer and Nystromformer limit the ability of looking at the full sequence and hence degrade long-range modeling capabilities. I do not see why PoNet is better than these low-rank methods on modeling long-range modeling."*
>
> It is challenging to perform robust low-rank approximations on a wide range of attention matrices, since the structure of the attention matrices varies. As compared in the LRA benchmark paper (Tay et al., 2021), low-rank methods may be less effective in hierarchically structured data and language modeling tasks. The goal of our work is to find a more efficient token mixing method for modeling token dependencies to replace the self-attention mechanism, rather than approximating self-attention such as low-rank methods. Hence, PoNet will not be constrained by the limitations of low-rank methods on modeling long-range dependencies. Our empirical results also showed that on LRA, PoNet achieves 61.05, outperforming Linformer (55.59) and Nystromformer (58.95).
>
> ### Re Question 1
>
> *"In the experiments of LRA, in Table 1 the authors said that they used the same setting as (Xiong et al., 2021). However, in Table 2 they said the setting is the same as (Tay et al., 2021). Are these two experiments using different settings? "*
>
> We apologize for the typo. Table 2 also used the same setting as (Xiong et al., 2021). We fixed this in the paper.
>
> ### Re Question 2
>
> *"In table 2, Performer is 4.3 times faster than Transformer under 4K sequence length. However, in the original LRA paper (Tay et al., 2021), the number is 5.7. What is the difference? "*
>
> We use the PyTorch codebase from(Xiong et al., 2021) for LRA experiments, which is different from the implementation in JAX/Flax for (Tay et al., 2021), so this implementation difference causes some difference in the results.
>
> ### Re Question 3
>
> *"In the LRA paper, the authors clarified that the sequences in the text classification task are trunked to length 4K. Why in Table 2, there are two columns with 8K and 16K lengths?"*
>
> The original LRA github also provides the full length datasets. Hence we could truncate the sequences in the text classification task into 8K and 16K lengths.
>
> ### Re missing references
>
> Thank you very much for pointing out these works. We have added them into related work, and also added discussions on comparisons between PoNet and Luna in Appendix D.3. Transformer-LS is quite different from our work, so we do not compare PoNet with it in detail in Appendix D.

---

> > ### Comment · Reviewer_Xvc7 · 2021-11-26
> > **Re: authors' feedback**
> >
> > We thank the authors for your feedback on our review.
> > After reading the authors' feedback, some of my concerns have been addressed.
> >
> > However, some (important) concerns are still there:
> >
> > 1. For the novelty of the global aggregation (GA), I still think it is similar to FastFormer, at least in the functional aspect. The differences between them lays in the instantialization details.
> >
> > 2. The more significant issues are from the experiments.
> >
> > a. To show the importance of SMP and LMP, the authors performed the ablation study in Table 5. However, this experiment of removing one or multiple components is not convincing, because removing components results smaller networks with less layers. A more convincing comparison is to replace the components with others to keep the network with similar size or layers.
> >
> > b. Due to different settings, only the results of the their section in Table 1 are comparable. To demonstrate the advantage of PoNet over low-rank approximation methods, more stronger baselines such as Luna is required.
> >
> > c. Thanks for clarifying the setting in Table 3. However, this clarification does not address my concerns but even makes it more salient. The models in the new Table 3 are pre-trained on a much smaller dataset (5GB) compared with BERT (16GB) and RoBERTa (160GB). We know that an important advantage of Transformer is its capacity. To fairly demonstrate the capacity of efficient Transformer architectures, we need to pre-trained the models on large corpus to visualize how much capacity we have to sacrifice to obtain efficiency. Pre-training on a small corpus cannot fully understand the capacity gap between different architectures. For example, in the Luna paper, when trained on 16GB data, Transformer does not show advantage over efficient architectures like Linformer or Luna. But when increasing the data size to 160GB, Transformer is still the best. It is highly recommended to conduct pre-training experiments on corpus of BERT and RoBERTa.

---

> > > ### Author Response · Authors · 2021-11-28
> > > **Re: Reviewer Xvc7 (1/4)**
> > >
> > > Thank you for all your feedback!
> > >
> > > Below we address all the concerns you raised and we further compare PoNet to Fastformer and Luna regarding your comments. Before comparing PoNet to Fastformer and Luna, we would like to gently remind that based on the ICLR2022 policy (which we cite below), both Fastformer and Luna works are considered contemporaneous to our PoNet work and our paper is not required to compare our work to these two works. Fastformer (Wu et al., 2021) was first posted on arXiv on August 20, 2021 (https://arxiv.org/abs/2108.09084) and it is still exclusively available on arXiv (not published at a peer-reviewed venue). Luna (Ma et al., 2021) was first posted on arXiv on Jun 3, 2021 (https://arxiv.org/abs/2106.01540), then accepted by NeurIPS 2021 which will start on December 6, 2021, and Luna’s camera-ready version was posted on arXiv on November 2, 2021.
> > >
> > > > https://iclr.cc/Conferences/2022/ReviewerGuide
> > > >
> > > > Q: Are authors expected to cite and compare with very recent work? What about non peer-reviewed (e.g., ArXiv) papers?
> > > >
> > > > A: We consider papers contemporaneous if they are published (available in online proceedings) within the last four months. That means, since our full paper deadline is October 5, if a paper was published (i.e., at a peer-reviewed venue) on or after June 5, 2021, authors are not required to compare their own work to that paper. Authors are encouraged to cite and discuss all relevant papers, but they may be excused for not knowing about papers not published in peer-reviewed conference proceedings or journals, which includes papers exclusively available on arXiv. Reviewers are encouraged to use their own good judgement and, if in doubt, discuss with their area chair.
> > >
> > > ### Re Concern 1
> > >
> > > *“For the novelty of the global aggregation (GA), I still think it is similar to FastFormer, at least in the functional aspect. The differences between them lays in the instantialization details.”*
> > >
> > > The instantiation details for the global aggregation (GA) module in our PoNet and Fastformer are different. As explained in our previous rebuttal, which is also summarized in Appendix D.1, our GA is different from Fastformer both in the motivation aspect and in the implementation aspect. Our GA is motivated by the global attention in Longformer and BigBird, having a set of global tokens that interact with all parts of the input sequence (PoNet uses one global token). And in the implementation aspect, the computation of the global query and also its interaction with the key and value are all different between GA and Fastformer.
> > >
> > > From the functional aspect, GA serves the purpose of modeling global contexts and then modeling the interactions between tokens and the global context representations. Fastformer also models global contexts, though differently through additive attention, and then models the interactions between tokens and the global context representations differently. From this very high level functional aspect, GA and Fastformer are similar, but so are Fastformer, BigBird, and Longformer.  In this general direction of modeling global contexts and interactions with tokens, there are different innovations on how to achieve it. Our paper demonstrates that GA as our approach of modeling global contexts and its interactions with tokens is novel (different from existing approaches) and is effective and efficient.
> > >
> > > As explained in the rebuttal, in addition to algorithm comparisons, we also implemented Fastformer and empirically compared the modeling capabilities between GA and Fastformer, and between PoNet and Fastformer. **We observe that PoNet demonstrates significantly better modeling capabilities compared to Fastformer, as follows.**
> > >
> > > **Replacing GA with Fastformer:**
> > >
> > > When replacing the GA module in our PoNet with Fastformer (and keeping SMP and LMP), we observe that after the same pre-training and fine-tuning configurations, replacing GA with Fastformer degrades performance compared to our PoNet: On CoLA, the accuracy drops **significantly** from PoNet’s $45.36$ to $43.28$; on STS-B, the accuracy also drops from $84.57$ to $84.46$.
> > >
> > > **Task accuracy of Fastformer pre-training:**
> > >
> > > When pre-training Fastformer and PoNet with the same data, MLM+SSO tasks, and configurations, **the MLM accuracy from Fastformer is lower than half of that from PoNet, and the SSO accuracy from Fastformer is also lower than that from PoNet.** Based on the GA only performance in Table 5, we hypothesize that the CoLA accuracy from Fastformer will be very low, consistent with our hypothesis that Fastformer does not sufficiently model local contexts and would be weak for single-sentence tasks.

---

> > > ### Author Response · Authors · 2021-11-28
> > > **Re: Reviewer Xvc7 (2/4)**
> > >
> > > ### Re Concern 2a
> > >
> > > *“To show the importance of SMP and LMP, the authors performed the ablation study in Table 5. However, this experiment of removing one or multiple components is not convincing, because removing components results smaller networks with less layers. A more convincing comparison is to replace the components with others to keep the network with similar size or layers.”*
> > >
> > > Thank you for the question. In the ablation study in Table 5, removing a module (GA, LMP, SMP) in PoNet means removing the dense linear transformations associated with it in the token-mixing layer. The table below shows the number of dense linear transformations for each model variant in Table 5. Note that PoNet has a total of 5 linear transformations (see Appendix F.1). Weakening the pre-training tasks to $L_{MN}$ and $L_{OM}$ does not change this number, i.e., still 5. The numbers of linear transformations for w/o SS-GA, w/o GA, w/o SMP, w/o LMP, w/o (SMP&LMP) are $4, 3, 4, 4, 3$, respectively. The number of model parameters for all model configurations in Table 5 is between $\boldsymbol{110M}$ and $\boldsymbol{124M}$, quite comparable to be considered as comparing models with similar sizes. Hence, we think the conclusions based on performance comparisons in Table 5 are valid.
> > >
> > > It is also important to point out that for PoNet, the same number of linear transformations indicates the same number of overall model parameters. Hence, w/o GA and w/o (SMP&LMP) have exactly the same number of model parameters. Based on Table 5, comparing these two rows, w/o (SMP&LMP) (i.e., GA only) degrades MLM accuracy significantly compared to w/o GA, from $56.64$ to $43.61$, causing CoLA accuracy to plunge from $49.51$ into $11.36$,  which proves that SMP and LMP are crucial for capturing local context information.  Although compared to w/o GA, w/o (SMP&LMP) improves SST accuracy and STS-B accuracy, overall on single-sentence and sentence-pair tasks,  w/o (SMP&LMP) degrades the performance of PoNet more than w/o GA. Results in Table 5 demonstrate the importance of SMP and LMP to the modeling capabilities of PoNet and the importance of all three poolings to the capabilities of PoNet.
> > >
> > > | Model                          | PoNet | w/o   SS-GA | w/o GA | w/o SMP | w/o LMP | w/o SMP&LMP | $L_{MN}$ | $L_{OM}$ |
> > > | ------------------------------ | ----- | ----------- | ------ | ------- | ------- | ----------- | -------- | -------- |
> > > | \#Dense Linear Transformations | 5     | 4           | 3      | 4       | 4       | 3           | 5        | 5        |

---

> > > ### Author Response · Authors · 2021-11-28
> > > **Re: Reviewer Xvc7 (3/4)**
> > >
> > > ### Re Concern 2b
> > >
> > > *“Due to different settings, only the results of the their section in Table 1 are comparable. To demonstrate the advantage of PoNet over low-rank approximation methods, more stronger baselines such as Luna is required.”*
> > >
> > > Yes, we compared PoNet with Luna on the LRA results on Page 7, in text in the updated draft. On Page 7, we state that “To the best of our knowledge, PoNet achieves very competitive accuracy on LRA against Transformer and recent efficient transformers, only lower than $63.09$ from AdaMRA (Zhang et al., 2021b), $61.95$ from Luna-256 (Ma et al., 2021), and $61.41$ from H-Transformer-1D (Zhu & Soricut, 2021).”
> > >
> > > Note that all results in the third group in Table 1, including baseline models and our PoNet, are fairly compared, since all models in this group are implemented based on the same PyTorch codebase based on (Xiong et al., 2021) and evaluated using the same configurations as used by (Xiong et al., 2021).
> > >
> > > **However, these results in the third group of Table 1 cannot be fairly compared to the LRA results from Luna (Ma et al., 2021), due to the following reasons.**  Firstly, we use Xiong et al. 's PyTorch codebase and Luna follows Tay et al.’s implementations (in JAX/Flax), which are not directly comparable. Hence, it is more reasonable to compare the gain over the respective vanilla Transformer results. Secondly, Luna did an adjustment to the LRA evaluation and obtained much better results ($59.24$) from their re-implemented Transformer over the original Transformer ($54.39$), as explained below in the Luna paper:
> > >
> > > > cite Luna: “For the task of Retrieval, we ﬁnd that models are not fully converged when being trained for 5K steps as stated in Tay et al. (2021). Therefore, we train models for 20K steps for this task and obtain much better results.”
> > >
> > > Since the authors apply this adjustment to Transformer (re-impl) and all Luna-L evaluations, it is reasonable to compare our PoNet’s gain over our Transformer baseline to Luna’s gain over their Transformer baseline. As shown in the third group in Table 1, our PoNet achieves $\boldsymbol{+2.28}$ gain ($61.05$ over $58.77$) on the average accuracy over the vanilla Transformer. In the Luna paper, their reimplemented Transformer,  Transformer (re-impl), achieves $59.24$ AVG, hence Luna-16 ($61.46$ AVG), Luna-128 ($61.93$), Luna-256 ($61.95$) achieve $+2.22$, $+2.69$, and $+2.71$ gain over their re-implemented Transformer, respectively.
> > >
> > > It is notable that comparing Table 2 in our paper and Table 2 in the Luna paper on training speed and peak memory consumption comparisons between different models, PoNet is significantly faster than Luna. With larger projected dimensions, i.e. 128 and 256, Luna requires more memory. When comparing PoNet with Luna-128 and Luna-256, PoNet is heavier than Luna for $1K-2K$ but already lighter in memory consumption than Luna for $>2K$ input sequence length. PoNet is significantly faster than Luna, with the training speed of PoNet reaching $\boldsymbol{4.4 \times}$ for $2K$ and $\boldsymbol{9.0 \times}$ for $4K$ input sequence length, whereas the speed of Luna-128 is $3.4\times$ for $2K$ and $5.1\times$ for $4K$ input sequence length.
> > >
> > > We also want to mention that regarding the GLUE results, the maximum sequence length of the GLUE tasks is around 128, while Luna uses 128 additional tokens as global tokens (Luna paper, Section 4.3), hence the amount of computations of Q, K, V are already consistent with that of Transformer, on top of the fact that Luna does cross-attention twice in the token-mixing layer.

---

> > > ### Author Response · Authors · 2021-11-28
> > > **Re: Reviewer Xvc7 (4/4)**
> > >
> > > ### Re Concern 2c
> > >
> > > *“Thanks for clarifying the setting in Table 3. However, this clarification does not address my concerns but even makes it more salient. The models in the new Table 3 are pre-trained on a much smaller dataset (5GB) compared with BERT (16GB) and RoBERTa (160GB). We know that an important advantage of Transformer is its capacity. To fairly demonstrate the capacity of efficient Transformer architectures, we need to pre-trained the models on large corpus to visualize how much capacity we have to sacrifice to obtain efficiency. Pre-training on a small corpus cannot fully understand the capacity gap between different architectures. For example, in the Luna paper, when trained on 16GB data, Transformer does not show advantage over efficient architectures like Linformer or Luna. But when increasing the data size to 160GB, Transformer is still the best. It is highly recommended to conduct pre-training experiments on corpus of BERT and RoBERTa. “*
> > >
> > > Thank you for your advice. Due to limited machine resources available to us and for achieving faster turnaround for the extensive experiments and ablation studies we conducted, we used the small Wikitext-103 (100M words) and Bookscorpus (800M words), in total \~5GB data size, for pre-training PoNet. We are in the process of further investigating the capacity of PoNet and pre-training PoNet with exactly the same pre-training data (16GB data) as used by BERT. It would be definitely useful to finish this experiment and we will report GLUE fine-tuning results once this experiment is completed.
> > >
> > > As to the concern on the tradeoff between model capacity and model efficiency and whether the capacity of Transformer is fairly demonstrated and compared in our paper, we want to emphasize that in the rebuttal, we have emphasized the importance of the two comparisons, Table 3 together with Figure 2, and **Table 7**. Table 3 provides a fair comparison between BERT, FNet, and PoNet, all Base-uncased models, when they are all pre-trained with the same 5GB pre-training data and same pre-training tasks and configurations and fine-tuned on GLUE with the same configurations.  Table 3 shows that the AVG score for BERT is $80.21$, $76.80$ for PoNet, and $73.46$ for FNet. We observe PoNet achieves **95.7%** of the accuracy of BERT on the GLUE benchmark, whereas FNet achieves only 91.6% of the accuracy of BERT on GLUE. These performance comparisons are consistent with the pre-training accuracies shown in Figure 2.
> > >
> > > **As to the concern that the capacity of Transformer requires a large amount of pre-training data to demonstrate, this is the purpose of the second group with (2) in Table 7, shown on Page 16 in Appendix B**.  The second group with (2) in Table 7 compared our PoNet pre-trained only on 5GB data to the official checkpoints of BERT and FNet. The official checkpoint of BERT was pre-trained on 16GB Wikipedia+Bookscorpus data, and that of FNet was pre-trained on 700GB C4 data, as in the original papers. Hence, **the capacity of both BERT and FNet models is considered to be fairly fully demonstrated.** Nevertheless, with the **much smaller pre-training data**, PoNet-Base still achieves GLUE AVG $77.54$, reaching **92.84%** of the accuracy of BERT on GLUE ($83.52$) and outperforms FNet ($74.23$) by **4.5%** relatively, and also better than $76.7$ reported in the original FNet paper (they used different code implemented with JAX/Flax).  Although StructBERT(Wang et al., 2019, Table 4) shows that the SSO objective can bring some improvement on BERT-Base performance over NSP on GLUE tasks, including CoLA (+0.8), MNLI (+1.1), QQP (+0.3), but no gain on SST-2,  the fact that PoNet, pre-trained on the much smaller 5GB data, can reach 92.84% of official BERT’s (pre-trained on 16GB data) accuracy on GLUE and outperform official FNet’s (pre-trained on 700GB data) accuracy by 4.5% relatively proves the promising potential of PoNet.  We are currently pre-training PoNet with the same 16GB pre-training data as BERT, and will update the GLUE results when they are ready.
> > >
> > > Thanks!

---

### Official Review · Reviewer_zcvh · 2021-11-04

**Correctness:** 3
**Technical Novelty And Significance:** 2
**Empirical Novelty And Significance:** 3
**Recommendation:** 8
**Confidence:** 4

**Main Review:**

The PoNet attention mechanism is very attractive as it intuitively makes sense with global, segment, and local inductive biases. PoNet also distinguishes itself with a low constant factor and performs well at even short sequence lengths. Experiments are very well done. Explanations and figures are clear.

I would have liked to see more discussion of the inductive biases beyond the ablation experiments and visualization in the appendix. For instance, I'd be curious if the different granularities are more or less important at different layers. Based on the single sentence in the appendix, it's hard to get an idea what the attention mechanism is actually doing, too. Some tasks beyond classification that may better be able to use structure like question-answering, translation, or summarization may be interesting, too.

**Summary Of The Paper:**

PoNet addresses the quadratic time and memory complexity of Transformer with a new attention mechanism that has only linear complexity. Theoretical (Section 3.2) and experimental results (Table 2) bear this out. Experimental results on standard datasets like Long Range Arena and GLUE are very competitive with SOTA. Ablations were done on the 3 components of the attention mechanism, too.

**Summary Of The Review:**

My recommendation is to accept the paper. It builds upon much work about exploiting inductive biases to make a more efficient transformer. While not revolutionary, it is a well-written paper with good analysis and experiments. The idea is has good intuition behind it, and the fact that it is a drop-in replacement for self-attention encoders makes it attractive for ML practicioners, too.

---

> ### Author Response · Authors · 2021-11-20
> **Response to Reviewer zcvh**
>
> Thank you very much for all the comments!
>
> First, we would like to raise a question on the Correctness score. The main claim of the paper is supported by our theoretical complexity analysis (Section 3.2) and empirical results and ablation analysis (Section 4 and Section 5). The Reviewer also stated in the review that "PoNet addresses the quadratic time and memory complexity of Transformer with a new attention mechanism that has only linear complexity. Theoretical (Section 3.2) and experimental results (Table 2) bear this out. Experimental results on standard datasets like Long Range Arena and GLUE are very competitive with SOTA. Ablations were done on the 3 components of the attention mechanism, too." We think the current score "**Correctness:** 1: The main claims of the paper are incorrect or not at all supported by theory or empirical results." is not justified and would like to check with the Reviewer on this score.
>
> Below we address all the comments from the Reviewer.
>
> ### **More Discussions on Inductive Biases**
>
> We updated Appendix to include C.1 for more discussions on inductive biases. To study the importance of the three granularities of pooling, i.e., GA, LMP, and SMP, at different layers of PoNet, we take 4000 examples, calculate the mean of the L2 Norm of each type of pooling for the 4000 examples at each layer, as well as the average of the means for the three types of pooling. We observe the following from Figure 3:
>
> 1. All three curves corresponding to the three types of pooling are close to the average curve in the graph, indicating that all three types of pooling play a significant role in PoNet. This observation is consistent with the ablation results shown in Table 5, which shows that removing GA, removing SMP, or removing LMP in PoNet all significantly degrade the performance on downstream single-sentence and sentence-pair tasks.
> 2. All three curves corresponding to GA, SMP, and LMP share a similar trend of going down, up, down, slowly rising and going down. From Layer 1 to 8, the norm values of SMP are greater than those of LMP and GA. Higher than Layer 9, the norm values of LMP turn out to be greater than those of SMP. These observations indicate that the significance of different granularities of pooling changes across layers.
> 3.  The norm value of GA is relatively low compared to that of SMP and LMP. On the other hand, as shown in the ablation results in Table 5, removing GA from PoNet (PoNet w/o GA) degrades performance on SST and STS-B significantly, showing that the sentence-pair tasks heavily rely on the global information.
>
> ### **More Tasks beyond Classification**
>
> We plan to investigate the performance of PoNet on other tasks including span-based question-answering tasks and generation tasks such as machine translation and summarization. For generation tasks, we plan to first evaluate using PoNet for encoder and the standard full self-attention Transformer decoder in the encoder-decoder framework on summarization tasks, similar to how Longformer and BigBird were evaluated on summarization tasks. This is based on the consideration that for long-form summarization tasks, the computational cost from full self-attention is more on the encoder, which needs to model long input sequence, whereas the decoder output is usually smaller, hence the full self-attention Transformer decoder can be used.For machine translation tasks on long-form input, the decoder output could also be long. To improve efficiency, we plan to investigate replacing self-attention in Transformer encoder with PoNet, and using PoNet for auto-regressive attention described in Appendix F.2 for decoder, respectively, as well as replacing encoder-decoder cross-attention with a pooling mechanism. Note that we describe in detail in Appendix F.2 on how to support auto-regressive attention in PoNet.
>
> Thanks!

---

### Author Response · Authors · 2021-11-22
**New Revision Uploaded**

We sincerely appreciate all the valuable comments and constructive suggestions from all the reviewers. All the reviews are tremendously helpful for us to improve the paper. We have uploaded a revised version of the paper to address these comments. We have posted detailed responses addressing all the weakness concerns and questions.

The major changes include:
1. Updated Table 3  and corresponding discussions in Section 4.2 for a fair comparison between BERT-Base, FNet-Base, and PoNet-Base on the GLUE validation set after we pre-train and fine-tune all these three models using the same configurations in pre-training and fine-tuning (details of pre-training in Appendix A.2 and details of fine-tuning in Appendix A.3). The fair comparison shows that PoNet achieves 95.7% of the accuracy of BERT on GLUE and outperforms FNet by 4.5% relatively.
2. Added Appendix B for additional GLUE results of comparing fine-tuning PoNet to fine-tuning official checkpoints of BERT and FNet and verification of correctness of our fine-tuning implementations.
3. Updated Appendix A.4 for clarifying the BERT-Base and FNet-Base results are from fine-tuning their official checkpoints.
4. Updated Table 5 with results from the PoNet GA-only variant (i.e., removing both SMP and LMP) and corresponding discussions to show the importance of SMP and LMP.
5. Added the number of parameters of PoNet in Section 4 with more details in the first paragraph of Appendix A.
6. Added more analysis on significance of different granularities of pooling at different layers of PoNet, in Appendix C.1.
7. Add two related work of Luna and Transformer-LS to Section 2 Related Work.
8. Added more details on comparing PoNet with Fastformer, Luna, Longformer, BigBird, and FNet in Appendix D, including performance from replacing the GA module in PoNet with Fastformer, and replacing the windowed local max-pooling in PoNet with random pooling similar to BigBird.
9. Added Appendix F to include ideas on how to further reduce the number of parameters of PoNet and how to support auto-regressive attention in PoNet.
10. Minor revisions for clarifying the comments from the reviewers.

---

### Decision · Program_Chairs · 2022-01-20

**Decision:**

Accept (Poster)

**Comment:**

This work reduces the time and memory complexity of Transformer for long sequences by using multiscale pooling to reduce attention from quadratic to linear complexity. Theoretical and experimental results show good results and are very competitive with the state-of-the-art. The paper is well written and experiments are thorough. The additional results in the rebuttal also helped reduce some of the reviewers' concerns. Though the work is somewhat incremental and the experimental settings for the baselines are different, the thoroughness of the experiments and the good results make this a good addition to the conference. For the final version, the authors should provide code, provide error bars and details of the speedup of GLUE.